# From the center of wind pressure to loads on the wind turbine: A stochastic approach for the reconstruction of load signals

Daniela Moreno <sup>1</sup>, Jan Friedrich <sup>1</sup>, Carsten Schubert <sup>2</sup>, Matthias Wächter <sup>1</sup>, Jörg Schwarte <sup>3</sup>, Gritt Pokriefke <sup>3</sup>, Günter Radons <sup>2,4,†</sup>, and Joachim Peinke <sup>1</sup>

**Correspondence:** Daniela Moreno (aura.daniela.moreno.mora@uni-oldenburg.de)

**Abstract.** In the context of the wind industry, there is an increasing need for a more comprehensive understanding of atmospheric wind conditions. A particular emphasis is required concerning wind structures, which have not been thoroughly investigated in the prevailing standard guidelines. This necessity arises in light of the current trends toward larger, higher, and more flexible wind turbine designs. Of particular importance are the correlations between the yet-to-be-characterized atmospheric turbulent structures and the specific responses of the turbines. These correlations may be crucial in assessing load events relevant to new designs that were negligible for the earlier, smaller, and stiffer turbines. The Center of Wind Pressure (CoWP) [Schubert et al., 2025] was recently introduced as a feature of a wind field that characterizes large-scale wind structures and, at the same time, correlates with the large-scale or low-frequency content of the bending moments at the main shaft of the wind turbines. In this paper, we comprehensively compare the CoWP and the bending moments in terms of their statistical properties and fatigue estimates, quantified by Damage Equivalent Loads (DEL). Furthermore, a stochastic method for the reconstruction of synthetic CoWP signals is proposed. The strong correlation with the bending moments enables the proposed stochastic CoWP model to serve as a relatively simple surrogate and estimator of the large-scale dynamics of these loads, which is based solely on the properties of the inflow wind field. A notable advantage of the stochastic approach is its capability to reconstruct very long time series, required for evaluating loads over the operational lifetime of the turbine. For such lifetime estimations on wind turbines, it is necessary to combine the proposed model for large-scale dynamics with a corresponding model for small-scale features, site-specific wind conditions, and turbine-specific characteristics. The proposed stochastic model of the CoWP can be used not only for load assessment, but also for characterizing large-scale wind structures. The model offers an advanced description of the wind phenomena, with the potential to be integrated as an extension of the prevailing wind turbulence models.

<sup>&</sup>lt;sup>1</sup>Carl von Ossietzky Universität Oldenburg, School of Mathematics and Science, Institute of Physics, Oldenburg, Germany.

<sup>&</sup>lt;sup>2</sup>ICM, Institute for Mechanical and Industrial Engineering Chemnitz.

<sup>&</sup>lt;sup>3</sup>Nordex Energy SE & Co.KG, Erich-Schlesinger-Straße 50, 18059 Rostock, Germany.

<sup>&</sup>lt;sup>4</sup>Institute of Physics, Chemnitz University of Technology, Germany.

<sup>†</sup>deceased, 20 July 2024

## 20 1 Introduction

As part of the design and validation phase, numerical simulations are used to predict the loads on an operational wind turbine (WT). The objective of these simulations is to reproduce the interaction between the WT and the atmospheric turbulent wind. Given the inherently complex meso-to-micro scale nature of atmospheric phenomena, it is extremely difficult to attempt to incorporate the governing physical models into a unified description of the wind flow. Consequently, stochastic wind models, which involve numerous simplifications and assumptions of the atmosphere, are commonly employed for numerical simulations of WTs. Common examples are the Kaimal (Kaimal et al., 1972), von-Karman (Von Kármán, 1948), and Mann (Mann, 1998) wind models. The International Electrotechnical Commission (IEC) (IEC, 2019) has proposed these models as standard atmospheric turbulence representations for numerical WTs simulations. It should be noted that these models are based on low-order statistical features of the wind fields, such as power spectra and correlations. However, they do not yet explicitly resolve the turbulent eddies, i.e., the spatial characteristics of the turbulent flow structures. For the spatial coherence of turbulence, an exponential decay with distance is assumed. The IEC standard also considers some extreme operating conditions (EOC), encompassing peak wind speeds, gusts, and sudden changes in wind direction. These non-realistic extreme wind structures are conceived as homogeneous in space (i.e., uniform over the entire rotor area), with a return period of 50 years.

Recent advancements in WT design show a persistent trend towards increasing dimensions, including higher heights and larger rotor diameters. Accordingly, certain structural properties are significantly modified within the designs of the larger WTs. Specifically, a higher degree of flexibility is characteristic of the larger and slimmer rotor blades. This may raise concerns about the validity of the assumptions or the omission of specific turbulent structures within the aforementioned standard wind models currently used by the WT industry. The increased scale of WTs suggests that certain wind characteristics, which were previously negligible or unimportant for smaller and more rigid WTs, may be significant considerations within the aerodynamic interactions of state-of-the-art WT designs. Of particular interest are the spatial properties of the atmospheric wind structures. Rotor diameters that exceed 200m may exhibit sensitivity to the spatial characteristics of wind phenomena, such as wind gusts.

The necessity for an *extended* characterization of the atmospheric turbulent wind beyond the parameters currently outlined in the IEC standard guidelines is supported by the repeated measurement of unexpected loads in operational WTs. According to manufacturers and operators of WTs, numerical simulations of the specific WTs and the standard IEC wind modeling assumptions do not adequately reflect certain load events that may be important for the structural integrity of the machines in operation. Consequently, it is imperative to establish a correlation between the extended features of the atmospheric wind and the measured unexpected effects on the operating WTs. Examples of such extended characteristics of atmospheric turbulence include the small-scale intermittency (Boettcher et al., 2003; Morales et al., 2012), low-level jets (Gutierrez et al., 2016), particular coherent vortices (Abraham and Hong, 2022), fractal turbulent-non-turbulent interfaces (Neuhaus et al., 2024), wind ramp events (Gallego-Castillo et al., 2015), and periods of constant wind speed (Moreno et al., 2025).

A general requirement within the wind industry is to simplify the complexity of WT representations in turbulent wind environments to allow practical implementation and minimize computational costs. As stipulated in the standard guidelines (IEC,

2019), numerical simulations of a wide range of operational scenarios are required for the validation of WT designs. Consequently, optimizing the computational time and power is imperative while ensuring satisfactory accuracy of the estimations of the responses of the WT. Some approaches have been proposed to reduce the complexity of the interaction between the wind and the WT. Examples of methods based on a given wind field include a modified actuator sector model for WT simulations (Mohammadi et al., 2024), and the calculation of extended equivalent wind speeds over the rotor area (Choukulkar et al., 2016). Conversely, techniques are employed to extract characteristics of the incoming flow field from load measurements at the WT, such as blade-load-based estimators (Coquelet et al., 2024). Furthermore, due to the limitations in computational power, the loads on the WT are typically estimated over short intervals, e.g., 10 minutes. Consequently, numerical techniques have been proposed for extrapolating the loads estimated from such short time scales to lifetime scenarios containing fatigue damage and extreme load events (Zhang and Dimitrov, 2023; Oingshan et al., 2022).

The virtual Center of Wind Pressure (CoWP) has recently been introduced as a feature of a given wind field that is either measured or modeled (Schubert et al., 2025). The CoWP characterizes large-scale wind structures occurring over the plane perpendicular to the main direction of the wind, i.e., the rotor plane, when considering a WT. Most interestingly, the CoWP is directly correlated to the low-frequency content of the bending moments at the main shaft of the WT. Consequently, the CoWP not only facilitates the characterization of extended wind structures, i.e., beyond the IEC standard, but also proposes a simplified and expeditious method for assessing particular characteristics of the WT loads.

In this article, we aim first to perform a comprehensive comparison between the CoWP, calculated from the wind fields, and the bending moments at the shaft of the WT, calculated using blade element momentum (BEM) numerical simulations. The statistical characteristics of the signals and their damage equivalent loads (DEL) are investigated. Second, based on the correlation between the large-scale structures of both the CoWP and the bending moments, we propose a stochastic method to derive the dynamics of the former, which are subsequently the basis for generating surrogate signals of the latter. The statistics of the surrogate data demonstrate a high degree of comparability to those of the original CoWP from the wind fields, as well as to the low-frequency content of the BEM-simulated bending moments. A notable advantage of the stochastic reconstruction is its capacity to generate very long time series. The availability of such extensive data is essential for assessing lifetime load events without the necessity of numerical extrapolation techniques.

Our model thus offers a twofold approach. On the one hand, it facilitates the characterization and modeling of large-scale wind structures. The wind energy sector is in urgent need of a comprehensive description of these large-scale structures, as standard wind models are likely to oversimplify them. Modern large wind turbines are particularly vulnerable to this oversimplification. On the other hand, our stochastic model allows the estimation and extrapolation of specific characteristics of the bending moments at the main shaft while bridging such responses of the WT with structures of the inflow wind field. In its current state, the method is limited to the modeling of the dynamics of the low-frequency components of the bending moments. However, when combined with a description of the high-frequency components, a validated rescaling procedure, and the characterization of the site-specific wind conditions, this approach enables a novel method for a fast assessment of the lifetime loads in WTs. In a preliminary investigation (Moreno et al., 2024), the stochastic method for reconstructing the time series of loads based on

the dynamics of the CoWP from IEC standard modeled wind fields was introduced. The present paper extends the stochastic approach to wind data from atmospheric measurements.

The paper is structured as follows: Sect. 2 presents the relevant definitions to be discussed in the paper. Sect. 3 describes the wind data to be investigated. The analysis of the reconstructed data from IEC standard-modeled wind fields and atmospheric measured data is presented in Sect. 4. Finally, the conclusions and outlook of our investigation are stated in Sect. 5.

# 2 Definitions

## 2.1 Center of Wind Pressure

The virtual Center of Wind Pressure (CoWP) is defined by Schubert et al. (2025) as the two-dimensional position in the plane of the rotor at which a point-wise thrust force  $F_T$  acts and induces the bending moments T. This position is specified with respect to a reference point, e.g., the main shaft of a WT. The moments are estimated as,  $T = CoWP \times F_T$ . Fig. 1 illustrates the concept of the CoWP, introduced as a characteristic of a given wind field  $\mathbf{u}(y,z,t)$ .

Figure 1. Schematic illustration of (a) the wind field  $\mathbf{u}(y,z,t)$  over the area of a rotor disk and (b) the resulting two-dimensional CoWP calculated from the wind field.

In the following, a brief derivation of the concept of the CoWP is presented. Let us consider a wind field  $\mathbf{u}(y_i, z_i, t)$  defined over a discretized grid with  $\mathcal{N}$  points on the rotor plane, i.e., the y-z plane, perpendicular to the main direction of the flow.

100 Then, the normal thrust force  $F_T$  acting over the rotor area  $\mathcal{A}$  at the y-z plane is calculated as,

$$F_T(t) = \frac{1}{2} \sum_{i=1}^{N} \rho_{air} C_T u^2(y_i, z_i, t) \Delta A_i$$
 (1)

where u is the longitudinal component of the wind perpendicular to the y-z plane,  $\rho_{air}$  is the density of air,  $C_T$  is the thrust coefficient of the rotor, and  $\Delta A_i$  are the discretized sections of the rotor area A. Now, the bending moments T due to the normal thrust force can be calculated as,

$$\mathbf{T}(t) = \tilde{\mathbf{r}}(t) \times F_T(t) \tag{2}$$

where  $\tilde{r}$  is the distance between the acting location of  $F_T$  to the reference point. Considering the main shaft, i.e., the center of the rotor disk  $(y_0, z_0)$  as the reference point, the yaw  $T_{yaw}$  and tilt  $T_{tilt}$  moments at the main shaft are estimated as,

$$T_{yaw}(t) = \frac{1}{2} \sum_{i=1}^{N} \tilde{y}_{i} \, \rho_{air} \, C_{T} \, u^{2}(y_{i}, z_{i}, t) \, \Delta \mathcal{A}_{i} \qquad \qquad T_{tilt}(t) = \frac{1}{2} \sum_{i=1}^{N} \tilde{z}_{i} \, \rho_{air} \, C_{T} \, u^{2}(y_{i}, z_{i}, t) \, \Delta \mathcal{A}_{i} \qquad (3)$$

where  $\tilde{y}$  and  $\tilde{z}$  are the horizontal and vertical distances, respectively, of each location i to the reference point, so that  $\tilde{y}_i = y_i - y_0$ 10 and  $\tilde{z}_i = z_i - z_0$ . Assuming  $\frac{1}{2} \rho_{air} C_T$  to be constant, the two CoWP components are defined as the fraction of the moments (yaw and tilt) and the normal thrust force, resulting in

$$CoWP_y(t) = \frac{\sum_{i=1}^{\mathcal{N}} \tilde{y}_i \ u^2(y_i, z_i, t) \ \Delta \mathcal{A}_i}{\sum_{i=1}^{n} u^2(y_i, z_i, t) \ \Delta \mathcal{A}_i} \qquad CoWP_z(t) = \frac{\sum_{i=1}^{\mathcal{N}} \tilde{z}_i \ u^2(y_i, z_i, t) \ \Delta \mathcal{A}_i}{\sum_{i=1}^{\mathcal{N}} u^2(y_i, z_i, t) \ \Delta \mathcal{A}_i}. \tag{4}$$

The CoWP is calculated solely by wind field data, comprehensively representing specific wind structures, whether modeled or measured. Furthermore, the area  $\mathcal{A}$  used to compute the CoWP can be adapted to investigate diverse sizes and domains within fields, e.g., 1D dynamics when measuring atmospheric data with vertically aligned devices in a met-mast, or different WT rotor sizes from numerically modeled wind fields.

## 2.2 Damage Equivalent Load

120

The Damage Equivalent Load (DEL) is the recommended method by the standard IEC (2019) for performing fatigue assessments and damage calculation analyses of the mechanical elements of the WT. In essence, the DEL represents a fixed amplitude and fixed-frequency load, calculated from a load signal encompassing a range of frequencies and amplitudes. Based on the Miner's rule (Miner, 1945), and the rainflow counting method (Matsuishi and Endo, 1968; Downing and Socie, 1982), the DEL is calculated over a period T as,

$$DEL = \left(\frac{\sum_{i=1}^{n} n_i s_i^m}{N_f}\right)_{T}^{m^{-1}} \quad , \tag{5}$$

where  $n_i$  is the number of cycles with amplitude  $s_i$ , and  $N_f$  is a reference number of cycles. The Wöhler exponent m is characteristic of the material, extracted from the so-called S-N curves (Basquin, 1910). Note that according to Eq. (5), the

contribution of the amplitudes  $s_i$ , to the DEL is determined by the exponent m. The larger the value of m, the stronger the dominance of larger amplitudes  $s_i$  within the calculation of the DEL. More details about the estimation and assumptions of the DEL can be found in (Sutherland, 1999). This study uses the DEL to evaluate the effect on the bending moments at the main shaft induced by the wind structures characterized by the CoWP.

# 130 2.3 The Stochastic Langevin Model

135

The CoWP calculated from a given wind field can be characterized in terms of its statistical properties and dynamical behavior. Since the CoWP signals are noisy and irregular, we introduce the Langevin model as a stochastic approach to characterize the dynamics. The range of applications of the Langevin method is extensive, encompassing domains as diverse as medical signals, e.g., the human balance (Rinn et al., 2016b; Bosek et al., 2004) or brain activity (Costa et al., 2016), financial markets (Friedrich et al., 2000), and cone penetration signals for stratigraphy (Lin et al., 2022).

Assuming a 1D stochastic process X(t), the general differential Langevin equation

$$\frac{d}{dt}X = D^{(1)}(X,t) + \sqrt{D^{(2)}(X,t)}\Gamma(t),\tag{6}$$

describes the temporal derivative  $\frac{dX}{dt}$  as the sum of two contributions: A deterministic part driven by the drift coefficient  $D^{(1)}$ , and a stochastic component driven by the diffusion coefficient  $D^{(2)}$  and weighted by a stochastic force  $\Gamma(t)$  (Lemons and Gythiel, 1997; Risken, 1996), where  $\Gamma(t)$  is Gaussian noise with zero mean and a  $\delta$ -correlation, i.e.,  $\langle \Gamma(t) \rangle = 0$ , and  $\langle \Gamma(t) \Gamma(t-t') \rangle = 2\delta(t-t')$ . The angular brackets  $\langle \dots \rangle$  denote the temporal average.

The Langevin method, introduced by Friedrich and Peinke (1997) and Siegert et al. (1998), proposes an approach to derive the coefficients  $D^{(k)}$  from time series X'(t). This is achieved by calculating the derivative of the conditional moments  $M^{(k)}(X,t,\tau)$  for the state X=X'(t) of the system as,

$$D^{(k)}(X,t) = \lim_{\tau \to 0} \frac{1}{\tau} M^{(k)}(X,t,\tau)$$
 (7)

for k=[1,2], where  $\tau$  is a small enough time step. The conditional moment  $M^{(k)}(X,t,\tau)$  is calculated by averaging the  $k^{th}$  power of the increments,  $X'(t+\tau)-X$ , as

$$M^{(k)}(X,t,\tau) = \frac{1}{k!} \langle [X'(t+\tau) - X]^k \big|_{X'(t) = X} \rangle.$$
(8)

Now, for a 2D process  $\mathbf{X}(\mathbf{t}) = \{X_1(t), X_2(t)\}$ , the Langevin equation has the form,

$$\frac{d}{dt} \begin{bmatrix} X_1 \\ X_2 \end{bmatrix} = \begin{bmatrix} D_1^{(1)}(\mathbf{X}, t) \\ D_2^{(1)}(\mathbf{X}, t) \end{bmatrix} + \begin{bmatrix} D_{11}^{(2)}(\mathbf{X}, t) & D_{12}^{(2)}(\mathbf{X}, t) \\ D_{21}^{(2)}(\mathbf{X}, t) & D_{22}^{(2)}(\mathbf{X}, t) \end{bmatrix} \begin{bmatrix} \Gamma_1(t) \\ \Gamma_2(t) \end{bmatrix},$$
 (9)

with the diffusion coefficients  $D_{12}^{(2)}$  and  $D_{21}^{(2)}$ ,

$$D_{12}^{(2)}(\mathbf{X},t) = D_{21}^{(2)}(\mathbf{X},t) = \frac{1}{2} \lim_{\tau \to 0} \frac{1}{\tau} \langle [X_1'(t+\tau) - X_1] [X_2'(t+\tau) - X_2] \big|_{X_1 = X_1'(t), X_2 = X_2'(t)} \rangle. \tag{10}$$

Once the coefficients  $D^{(1,2)}$  are known, the method can be reversed. Then, time series X(t) can be generated via the stochastic integration of Eq. (9). The application of the Langevin approach for the stochastic reconstruction of the time series of the two-dimensional CoWP is discussed in the next section. Further developments and details on the Langevin model are found in Friedrich et al. (2011); Reinke et al. (2015); Rinn et al. (2016a); Tabar (2019).

# 2.4 Stochastic Model for CoWP and WT Loads

170

In Fig. 2, we schematically show our proposed stochastic method in the context of load estimation of the low-frequency contribution of the bending moments at the main shaft of a WT. Starting from either a modeled or measured wind field u(y,z,t), the CoWP is calculated according to Eq. (4) (going in the upward direction in Fig. 2). Then, the stochastic Langevin approach is used to derive the coefficients  $D^{(1,2)}$  from the CoWP signals. Based on the extracted coefficients  $D^{(1,2)}$ , the stochastic reconstruction of signals of the low-frequency bending moments at the main shaft can be achieved. The strength of this approach lies in its ability, based on the Langevin stochastic differential equation, to generate a time series of any length while preserving the statistical properties of the original CoWP data from the wind field data. This feature is particularly advantageous, as lifetime load assessments of a WT require large amounts of computationally expensive data or numerical extrapolation techniques. However, the implementation of the stochastic method for lifetime load assessment in engineering applications (i.e., including both the high- and low-frequency components of the loads) is constrained in its application. To ensure a comprehensive lifetime model, it is necessary to incorporate additional elements. A turbine-specific transfer function for rescaling the magnitudes of the loads is required. A complementary model for the high-frequency components of the loads must be integrated. Finally, the occurrence of the loads must be weighted by the distribution of the mean wind speed at the location of the WT (e.g., annual Weibull distribution).

The current standard procedure for load assessment in the wind industry is shown in the downward direction from the wind field  $\mathbf{u}(y,z,t)$  in Fig. 2. In brief, the response of a WT to a specific inflow wind field is investigated via a BEM simulation, with a typical length of 10 minutes. The time series of the loads are obtained from several 10-minute random realizations that account for different wind conditions. Thus, the assessment of all required wind situations is computationally very demanding. After the aggregation of 10-minute BEM simulations, extrapolation methods are applied to account for extreme load events and damage calculation during the lifetime of the WT (Zhang and Dimitrov, 2023; Qingshan et al., 2022). Compared to the standard approach, our proposed model is computationally very efficient and thus fast. The lowest path in Fig. 2, depicted by dashed lines, shows the potential use of operational wind and load measurements for validation and optimization processes.

As a side comment, the description of the dynamics of the CoWP, e.g., via the derivation of  $D^{(1,2)}$ , provides a comprehensive characterization of the large-scale structures in the wind field, which can be further investigated. For instance, it can be used to estimate the accuracy of modeled wind data compared to atmospheric measurements, or it can be included as a parametrization into extensive descriptions of the turbulent wind, such as the IEC standard models or other surrogate models for wind field reconstruction (Yassin et al., 2023; Friedrich et al., 2022; Rinker, 2018).

Figure 2. Diagram of the stochastic surrogate method for the assessment of the low-frequency content of the bending moments at the main shaft given a wind field  $\mathbf{u}(x,y,z)$ . The proposed method goes upwards from the wind field in the figure. For comparison, the path in the downward direction shows the standard procedure for load estimations specified by the IEC standard guideline. The solid red box contains the three data sets to be compared in the following sections. The dashed lines at the lowest part of the figure represent operational measured data to be potentially included in an extended comparison.

- The solid-line red box in Fig. 2 shows schematically the three data sets to be compared and discussed in the following sections of this paper. From bottom to top:
  - a) the bending moments at the main shaft calculated by BEM simulations.
  - b) the CoWP calculated from the modeled or measured wind fields (see Sect. 3).
  - c) the time series generated via the stochastic reconstruction.

190 Again, the dashed line shows a potential use of operational load data to be included in the comparison.

# 3 Wind Data: IEC Standard Fields and Atmospheric Measurements

We aim to characterize and model the CoWP from two wind data sets:

i) IEC standard Kaimal data: Synthetic wind fields are generated with the Kaimal model (Kaimal et al., 1972) proposed by the IEC standard(IEC, 2019). The fields are defined in a  $130\,\mathrm{m}\,\mathrm{x}\,130\,\mathrm{m}$  spatial grid with a separation  $\Delta y = \Delta z = 10\,\mathrm{m}$  and centered at y = 0 and  $z = 90\,\mathrm{m}$ . The grid points are schematically shown by the small black dots in Fig. 3. The circular gray area depicts the scaled rotor of the 5MW NREL turbine (Jonkman et al., 2009) with hub height at  $90\,\mathrm{m}$  and a rotor diameter of  $126\,\mathrm{m}$ . The mean wind speed  $\bar{u} = 7\,\mathrm{m}\,\mathrm{s}^{-1}$ , and turbulence intensity  $\mathrm{TI} = 7\%$  of the non-shear wind fields are defined at the location of the hub. The BEM simulations of the 5MW NREL turbine are performed in OpenFAST (Jonkman et al.).  $4.7\mathrm{x}10^4\,\mathrm{s}$  of simulated time is investigated. The TurbSim Package (Jonkman, 2016) was used to generate the Kaimal fields. The implementation in TurbSim of the Kaimal spectrum for the longitudinal component u of the wind follows.

$$E_u(f) = \frac{4\sigma_u^2 L_u/\bar{u}_H}{(1 + 6fL_u/\bar{u}_H)^{5/3}}$$
(11)

where  $\sigma_u$  is the standard deviation of u,  $\bar{u}_H$  is the mean at hub height, and f is the frequency. The integral scale  $L_u$  is defined as  $L_u = 8.10\Lambda_u$ , with  $\Lambda_u$  being the turbulence scale.  $\Lambda_u$  is calculated as  $\Lambda_u = 0.7 \, (\min\{30\text{m}, H_H\})$ , where  $H_H$  is the hub height. In conjunction with the Kaimal spectra, an exponential coherence model is assumed to describe the spatial correlation of the longitudinal component u. The coherence scale parameter  $L_c$  for the coherence model(IEC, 2019) is assumed as  $L_c = L_u = 8.10\Lambda_u$ .

Figure 3. Schematic representation of the Kaimal spatial grid and the 5MW NREL rotor. The black points depict the discrete locations of the spatial grid with  $\Delta y = \Delta z = 10$  m. The hub of the model WT is located at the center point of the grid at y = 0 m and z = 90 m.

ii) Atmospheric GROWIAN data: The measurement campaign was conducted in Germany between 1984 and 1987. The horizontal wind speed was measured with a frequency of  $2.5\,\mathrm{Hz}$  by  $16\,\mathrm{propeller}$  anemometers arranged in two met masts, covering an area of  $76\,\mathrm{m}\,\mathrm{x}\,100\,\mathrm{m}$ . Details of the GROWIAN data are found in (Körber et al., 1988; Günther and Hennemuth, 1998). The blue circles in Fig. 4 illustrate a schematic representation of the measurement arrangement. The GROWIAN data have been conditioned by the mean wind speed  $8.5 \le \bar{u} 

**Figure 4.** Schematic representation of the GROWIAN spatial grid. The blue circles show the original GROWIAN arrangement. The green circles show the locations of the stretched grid used for BEM simulations of a WT. The gray area depicts the WT model to be simulated with hub height at 125 m and rotor diameter of 149 m.

It is important to note that the simplified wind field rescaling in this study results in some degree of distortion to the spatial correlations of the original GROWIAN data. The introduced distortion, under the assumption of self-similarity of turbulence, is of minor significance, as our primary utilization of the GROWIAN data is to develop realistic, large-scale structures of the turbulent atmospheric boundary layer. Large-scale wind structures (e.g., of the size of the rotor diameter) are not present in other standard numerical wind fields. They will become important for the CoWP and the loads, as demonstrated subsequently.

## 4 Results and Discussion

230

235

240

This section presents the results of the two objectives of our investigation: The comparisons between the CoWP and the bending moments at the main shaft of the WT, and the stochastic modeling of the CoWP. In Sect. 4.1, the results from the standard

modeled Kaimal data are shown. Respectively, in Sect. 4.2, the investigation of the atmospheric GROWIAN measurements is presented. The analysis of the two data sets is performed as follows: The two components of the CoWP=  $\{CoWP_y, CoWP_z\}$  are calculated from the wind fields according to Eq.(4), with the hub location, i.e., equivalent to the location of the main shaft, as the reference point  $(y_0, z_0)$ . The Langevin stochastic approach described in Sect. 2.3 is then applied for modeling random signals of the CoWP. The characteristics of the original CoWP, the modeled CoWP, and the BEM simulated bending moments  $T = \{T_{\text{yaw}}, T_{\text{tilt}}\}$  at the main shaft of the WT are compared. For the comparison, the statistics over time, as well as the DEL, are analyzed.

In Schubert et al. (2025) it has been shown that the CoWP can be used as a description of wind structures with temporal scales larger than  $10 \, \text{s}$ . Accordingly, the correlation to the bending moments is limited to the low-frequency component. Therefore, to discard the high-frequency content, the signals are low-pass filtered. This applies to both the bending moments and the CoWP. The filter is a finite impulse response (FIR) filter with the cutoff or pass-band frequency  $f_{cutoff}$ . The value of  $f_{cutoff}$  should be lower than the rotational frequency P of the WT. In this way, the effect of gravitational loads from the rotating blades (i.e., P and 3P) is averaged out. Here, a  $0.1 \, \text{Hz}$  cutoff frequency is applied. For comparability, the signals have also been normalized to have a zero mean and a standard deviation equal to 1. The comparisons presented in the following sections are made using the signals of the CoWP and the bending moments after frequency filtering and normalization.

## 4.1 The CoWP from standard Kaimal wind fields

250

255

260

265

270

# The CoWP and the bending moments at the main shaft

We start by comparing the CoWP calculated from the Kaimal wind fields and the bending moments T from the BEM simulations. Fig. 5 shows 20-min excerpts of the time series of the CoWP and the bending moments. In (a) CoWP $_y$  and  $T_{yaw}$ , and in (b) CoWP $_z$  and  $T_{tilt}$  are shown. The observed correlation between the time series of the CoWP and the bending moments is quantified in Fig. 6. Each time step in the time series is represented by a point (CoWP(t), T(t)). In (a)  $T_{yaw}$  and CoWP $_y$ , and (b)  $T_{tilt}$  and CoWP $_z$ . The observed linear behavior with a slope of approximately 0.93 quantifies the strong correlation between the normalized CoWP, which characterizes particular structures within the wind field, and the normalized bending moments experienced by the WT interacting with such a wind field. These correlations obtained from the standard Kaimal wind field corroborate the findings presented in (Schubert et al., 2025), where the CoWP calculated from atmospheric measured data demonstrated correlation coefficients up to 0.9 with the corresponding BEM-simulated yaw and tilt bending moments at the main shaft. A turbine-specific transfer function for rescaling the normalized values of the CoWP to magnitudes of the low-frequency component of operational bending moments would be necessary for the assessment of the loads in engineering applications. Such a transfer function will therefore depend on the structural properties of the WT and particular control mechanisms.

Figure 5. 20-min excerpts of the CoWP and the bending moments at the main shaft of a WT. (a)  $T_{\text{yaw}}$  and  $CoWP_y$ , and (b)  $T_{\text{tilt}}$  and  $CoWP_z$ . The signals are normalized and low-pass filtered.

Figure 6. CoWP against the bending moments plotted as (CoWP(t), T(t)) for each time step t of the time series. In (a)  $T_{\text{yaw}}$  and  $CoWP_y$ , and (b)  $T_{\text{tilt}}$  and  $CoWP_z$ . The gray lines depict linear fittings T = a (CoWP) + b. The values of the Root Mean Square Error (RMSE) are shown in the legends. The signals are normalized and low-pass filtered.

As a further statistical comparison Fig. 7 shows the probability density functions (PDF) of the CoWP and of the bending moments taking into account all the simulated data. It should be noted that the rare large events, depicted by the tails of the PDFs are in good agreement.

Figure 7. PDF of the CoWP and the bending moments at the main shaft of a WT. (a)  $T_{\text{yaw}}$  and CoWP<sub>y</sub>, and (b)  $T_{\text{tilt}}$  and CoWP<sub>z</sub>. The signals are normalized and low-pass filtered.

Now that we have proven the strong correlation between the dynamics of the low-frequency content of the CoWP and the bending moments, we continue by introducing the  $DEL_{CoWP}$ . The  $DEL_{CoWP}$  follows from Eq. (5) as

$$DEL_{CoWP} = \left(\frac{\sum_{i=1}^{n} (n_{i,CoWP} s_{i,CoWP}^{m})}{N_f}\right)_{T}^{m-1} , \qquad (12)$$

where the number of cycles  $n_{i,\text{CoWP}}$ , and the amplitudes  $s_{i,\text{CoWP}}$  are derived from the CoWP signals.

In Schubert et al. (2025), it was demonstrated that high values of the DEL are driven by significantly large amplitude events in the low-passed filtered time series of the loads. Additional proof for this correspondence is given in Appendix A. Accordingly, large amplitude events in the signals of the CoWP will result in high values of a DEL<sub>CoWP</sub>.

A good agreement between the DEL and the DEL<sub>CoWP</sub> implies that estimations of the low-frequency events of the bending moments at the main shaft of a WT can be accomplished purely from the estimation of the CoWP from the incoming wind field. Fig. 8 (a) and (b) show the correlation plots of the resulting time-resolved DEL and DEL<sub>CoWP</sub> obtained through an averaging of time  $T = 60 \, \text{s}$  and a coefficient m = 10. Their statistics are summarized in the box plots in (c) and (d). The DEL and DEL<sub>CoWP</sub> are calculated in (a) and (c) from  $CoWP_y$  and  $T_{yaw}$ , and in (b) from  $CoWP_z$  and  $T_{tilt}$ . A lower correlation is obtained for the DEL and  $DEL_{CoWP}$  in the vertical direction in panel (b) compared to the horizontal component shown in (a). The lower correlation is explained by the more scattered results within the correlation of the time series of the  $T_{tilt}$  and the  $CoWP_z$  shown in Fig. 6. There, a value of RSME = 0.40 indicates a higher degree of scattering for  $T_{tilt}$ , compared to a RSME = 0.34 for  $T_{yaw}$ . Overall, the data in Fig. 8 reveal a very good agreement between the DEL and  $DEL_{CoWP}$  in a statistical sense. Although a spread of the data is observed, the statistics and correlation are consistent. In an aggregate sense, these results indicate an equivalence between the CoWP and the bending moments. The validity of the method has been proven for the rated power regime of the WT.

Figure 8. Comparison between the DEL and DEL<sub>CoWP</sub>. Correlation plots and box plots for CoWP<sub>y</sub> and  $T_{\text{yaw}}$  in (a) and (c), and CoWP<sub>z</sub> and  $T_{\text{tilt}}$  in (b) and (d). The gray lines in (a) and (b) depict linear fittings. The values of the RMSE are shown in the legends. In the box plots in (c) and (d), the horizontal line inside each box shows the median, and the bottom and top edges indicate the 25th ( $P_{25}$ ) and 75th ( $P_{75}$ ) percentiles. The whiskers indicate the most extreme data points. They are calculates as  $P_{25} - (1.5 \times \text{IQR})$  and  $P_{75} + (1.5 \times \text{IQR})$ , where IQR is the interquantile range IQR =  $P_{75} - P_{25}$ . The markers show outliers. The DEL and DEL<sub>CoWP</sub> are calculated with m = 10 over periods T = 60 s with 30 s overlapping between two consecutive periods. The signals are normalized and low-pass filtered.

It is essential to acknowledge that the discussion on the DELs presented in our work is exclusively focused on the DELs from the low-frequency component of the signals. This choice is based on a particular interest of our research partners. In order to assess the complete DELs (e.g., from both the low- and high-frequency load events), it is necessary to establish an additional model for incorporating the contribution from the high-frequency signal. In this direction, a simple surrogate stochastic model has shown satisfactory results. The characteristics of the original high-frequency load signal are well reproduced. The proposed stochastic model for the high-frequency signals, and calculations on the differences between the DELs from the low- and high-frequency load components, and total DELs are shown in Appendix B.

# 300 Stochastic reconstruction of the CoWP

We now apply the stochastic Langevin approach introduced in Sect. 2.3 as a method for characterizing the low-frequency dynamics of the CoWP from the Kaimal wind fields. The 2D stochastic differential equations (see Eq. (9)) are thus applied for  $CoWP(t) = \{CoWP_y(t), CoWP_z(t)\}$ . Since the two components  $CoWP_y(t)$  and  $CoWP_y(t)$  proved to be uncorrelated, i.e.,  $D_{12}^{(2)} = D_{21}^{(2)} = 0$ , the coefficients  $D^{(1,2)}$  are independently estimated from the time series of  $CoWP_y$  and  $CoWP_z$  according to Eqs. (7) and (8). The results on the calculation of the correlation function  $\langle CoWP_i(t) CoWP_j(t+\tau) \rangle$  for i=1,2 are shown in Appendix C).

The results of the coefficients  $D^{(1,2)}$  are shown in Fig. 9. The linear dependence of the drift coefficients  $D^{(1)}$  in (a) is clear for the two components CoWP<sub>y</sub> and CoWP<sub>z</sub>. An almost constant diffusion term  $D^{(2)}$  is observed in (b) for the two components.

Figure 9. Langevin approach of the CoWP from Kaimal wind fields. (a) Drift coefficient  $D^{(1)}$ , and (b) Diffusion coefficient  $D^{(2)}$ . In black for the vertical component CoWP<sub>y</sub>, and in red for the horizontal component CoWP<sub>z</sub>.

The estimated  $D^{(1)}$  and  $D^{(2)}$  are used for the reconstruction of synthetic time series (CoWP<sub>R</sub>) via the stochastic integration of Eq. (9). A signal CoWP<sub>R</sub> with a length of  $4.7x10^4$  s is reconstructed.

For a first visual comparison between the original and the reconstructed signal, the filtered but non-normalized trajectories of the CoWP and CoWP<sub>R</sub> in the y-z plane are shown in Fig. 10. Symmetric paths, i.e., comparable magnitudes in the two directions y and z, are observed for CoWP and CoWP<sub>R</sub>.

Figure 10. Trajectories on the y-z plane of (a) the original CoWP calculated from the Kaimal data, and (b) the stochastically reconstructed signal CoWP $_R$ . Intentionally, the data of both CoWP and CoWP $_R$  for plotting the trajectories in (a) and (b) are not normalized.

Note that due to the stochastic reconstruction, temporal correlation is not expected between the two signals. However, a statistical similarity should be present. This is shown in Fig. 11, which compares the PDF of the signals. After filtering and normalization, the results of the BEM-simulated bending moments at the main shaft are also included. In (a), the components in the horizontal *y* direction. In (b), the components in the vertical *z* direction.

320

Figure 11. PDF of the CoWP, CoWP<sub>R</sub> and bending moments T. (a) CoWP<sub>R,y</sub>, CoWP<sub>y</sub>, and  $T_{yaw}$ . (b) CoWP<sub>R,z</sub>, CoWP<sub>z</sub>, and  $T_{tilt}$ . The signals are normalized and low-pass filtered.

To characterize in more detail the similarity of the signals, we also investigate the statistics of their *increments* or their variations for a given time scale  $\tau$ . The increments are defined as  $\Delta x_{\tau}(t) = x(t+\tau) - x(t)$ , for a given signal x(t) and include two-time correlations like the autocorrelation or the power spectrum (Morales et al., 2012). Fig. 12 shows the excellent accordance of

the increments statistics of  $\Delta \text{CoWP}_{\tau}$ ,  $\Delta \text{CoWP}_{R,\tau}$  and  $\Delta T_{\tau}$  for values of  $\tau = [5, 10, 20, 30] \text{s}$ . In the upper row, the components in the horizontal y direction and in the lower row, the components in the vertical z direction are shown.

Figure 12. PDF of the increments with  $\tau = [5, 10, 20, 30]$ s. (a) upper row, horizontal component:  $\Delta \text{CoWP}_{y,\tau}$ ,  $\Delta \text{CoWP}_{R,y,\tau}$  and  $\Delta T_{yaw,\tau}$ . (b) lower row vertical component:  $\Delta \text{CoWP}_{z,\tau}$ ,  $\Delta \text{CoWP}_{R,z,\tau}$  and  $\Delta T_{tilt,\tau}$ . The time series are normalized and low-pass filtered.

Finally, we show in Fig. 13 the accordance of the resulting DEL, DEL<sub>CoWP</sub> and DEL<sub>CoWPR</sub> by box plots. A subindex R refers to the reconstructed signal ((a) the components in the horizontal y direction; (b) the components in the vertical z direction). As observed from the box plots, the distributions of the DEL<sub>CoWPR</sub> from the stochastically reconstructed signal CoWPR reproduce quite accurately the distributions of both the DEL<sub>CoWP</sub> from the original CoWP and the DEL from the BEM simulated signals.

Figure 13. Box plots of the DEL<sub>CoWP</sub>, DEL, and DEL<sub>CoWPR</sub> from normalized and filtered signals of (a) CoWP<sub>y</sub>,  $T_{\text{yaw}}$ , and CoWP<sub>R,y</sub>, and (b) CoWP<sub>z</sub> and  $T_{\text{tilt}}$ , and CoWP<sub>R,z</sub>. The DELs are calculated over periods T = 60 s with 30 s of overlapping and with a coefficient m = 10. The lines defining each box show the median, and the bottom and top edges indicate the 25 th ( $P_{25}$ ) and 75 th( $P_{75}$ ) percentiles. The whiskers indicate the most extreme data points. They are calculates as  $P_{25} - (1.5 \times \text{IQR})$  and  $P_{75} + (1.5 \times \text{IQR})$ . The markers show outliers.

## 4.2 The CoWP from atmospheric GROWIAN measurements

Next, we use the atmospheric GROWIAN wind fields described in Sect. 3 to calculate the CoWP, to simulate the BEM bending moments at the main shaft, and to apply the stochastic Langevin model for the reconstruction of random data. The results are presented in the same sequence as done for the Kaimal data in the previous section.

## The CoWP and the bending moments at the main shaft

Fig. 14 shows the correlation plots between the CoWP and the bending moments T. In (a)  $T_{\rm yaw}$  and CoWP $_y$ , and (b)  $T_{\rm tilt}$  and CoWP $_z$ . The coefficients of the linear fittings agree with the correlation coefficients of around 0.9 reported in Schubert et al. (2025), where all the available GROWIAN wind fields are investigated. Differently, in this paper, we only investigate a subset of the atmospheric data, conditioned by the mean wind speed, turbulence intensity, and shear exponent, as described in Sect. 3. The correlation between the CoWP and the bending moments is slightly decreased compared to the standard modeled wind data shown in Fig. 6. In particular, *loops* are observed in Fig. 14 for the two components, (a) and (b). Such loops correspond to intervals where more severe wind conditions affect the WT than we find in the Kaimal wind data. Over those intervals, significant differences in the wind speed are observed in the spatial domain (i.e., over the rotor plane). As a result, divergences in calculating the CoWP and the bending moments are obtained. Examples of such severe wind conditions within the atmospheric rescaled GROWIAN fields are shown in Appendix D.

Figure 14. CoWP against the bending moments T at the main shaft of a WT plotted as (CoWP(t), T(t)) for each time step t of the time series. In (a)  $T_{yaw}$  and  $CoWP_y$ , and (b)  $T_{tilt}$  and  $CoWP_z$ . The gray lines depict linear fittings. The values of the RMSE are shown in the legends. The time series are normalized and low-pass filtered.

# Reconstructing the CoWP from atmospheric wind data

The results of the coefficients  $D^{(1,2)}$  from the stochastic Langevin method applied to the CoWP from the GROWIAN measurements are shown in Fig. 15. Interestingly, the diffusion coefficients  $D^{(2)}$  in (b) are clearly not constant. This behavior is called multiplicative noise and is significantly stronger for the vertical component  $CoWP_z$ . In contrast, pure additive noise was obtained for the modeled Kaimal fields in Fig. 9. Moreover, the diffusion coefficient  $D^{(2)}$  of the  $CoWP_z$  from Kaimal data with shear, showed pure additive noise (see Appendix E). This effect in  $D^{(2)}$  is a consequence of the different wind fields and shows that the Kaimal data lead to simpler noise. In contrast, atmospheric wind data have more complicated deterministic and noise contributions.

**Figure 15.** Langevin approach of the CoWP from GROWIAN wind fields. (a) Drift coefficient  $D^{(1)}$ , and (b) Diffusion coefficient  $D^{(2)}$ . In green for the vertical component CoWP<sub>u</sub>, and red for the horizontal component CoWP<sub>z</sub>.

Fig. 16 shows the trajectories of the CoWP and the CoWP<sub>R</sub> in the y-z plane. For this representation, the time series are not normalized. Due to the shear, the movement of the CoWP in the vertical direction z is larger than in the horizontal direction y. This differs from Fig. 10, where symmetric trajectories in the two directions y-z are obtained for non-shear Kaimal wind fields.

Figure 16. Trajectories on the y-z plane of (a) the CoWP from the original GROWIAN measurements, and (b) the stochastically reconstructed signal CoWP $_R$ . The data are not normalized.

Fig. 17 shows the PDF of the signals. The time series of the BEM simulated bending moments  $T_{yaw}$  and  $T_{tilt}$  are also included. 355 In (a) for the horizontal y component, and in (b) for the vertical z component. We see that the stochastic model reproduces the statistics of the CoWP and bending moment very well. The PDFs of Fig. 17 show additional structures like skewness and small bumps. These structures are the consequence of the nonlinearities of  $D^{(1,2)}$  in Fig. 9, (see also the stationary solution of the Fokker-Planck equation which corresponds to the Langevin equation (Risken, 1996)).

**Figure 17.** PDF of the the CoWP, CoWP<sub>R</sub> and bending moments T. In (a) CoWP<sub>y</sub>,  $T_{\text{yaw}}$ , and CoWP<sub>R,y</sub>. In (b) CoWP<sub>z</sub> and  $T_{\text{tilt}}$ , and CoWP<sub>R,z</sub>. The signals are normalized and low-pass filtered.

As higher order statistical feature in Fig. 18 the PDFs of the increments  $\Delta \text{CoWP}_{\tau}$ ,  $\Delta \text{CoWP}_{R,\tau}$  and  $\Delta T_{\tau}$  for values of  $\tau = [5, 10, 20, 30]$ s are shown ((a) the components in the horizontal y direction; (b) the components in the vertical z direction).

Figure 18. PDF of the increments with  $\tau = [5, 10, 20, 30]$ s. (a) Horizontal component:  $\Delta \text{CoWP}_{y,\tau}$ ,  $\Delta \text{CoWP}_{R,y,\tau}$  and  $\Delta T_{yaw,\tau}$ . (b) Vertical component:  $\Delta \text{CoWP}_{z,\tau}$ ,  $\Delta \text{CoWP}_{R,z,\tau}$  and  $\Delta T_{tilt,\tau}$ . The time series are normalized and low-pass filtered.

Finally, Fig. 19 (a) and (b) show the correlation plots of the DEL and DEL<sub>CoWP</sub>. In (c) and (d) the box plots describing their statistics are shown. The box plots of the DEL<sub>CoWPR</sub> in the two components are also included. The DEL and DEL<sub>CoWPR</sub> are calculated in (a) and (c) from CoWP<sub>y</sub> and  $T_{yaw}$ , and in (b) from CoWP<sub>z</sub> and  $T_{tilt}$ . All time series are normalized and filtered.

Figure 19. Comparison between the  $\mathrm{DEL_{CoWP}}$ ,  $\mathrm{DEL}$ , and  $\mathrm{DEL_{CoWP}}_R$ . Correlation plots and box plots for  $\mathrm{CoWP}_y$  and  $T_{\mathrm{yaw}}$  in (a) and (c), and  $\mathrm{CoWP}_z$  and  $T_{\mathrm{tilt}}$  in (b) and (d). The gray lines in (a) and (b) depict linear fittings. The values of the RMSE are shown in the legends. In the box plots in (c) and (d), the horizontal line inside each box shows the median, and the bottom and top edges indicate the 25th  $(P_{25})$  and 75th  $(P_{75})$  percentiles. The whiskers indicate the most extreme data points. They are calculates as  $P_{25} - (1.5 \times \mathrm{IQR})$  and  $P_{75} + (1.5 \times \mathrm{IQR})$ . The markers show outliers. The DELs are calculated with m=10 over periods  $T=60\,\mathrm{s}$  with  $30\,\mathrm{s}$  overlapping between two consecutive periods. The signals are normalized and low-pass filtered.

The correlations between the low-passed and normalized signals of the CoWP and the bending moments shown in Fig. 14, and between the DEL $_{CoWP}$  and DEL shown in Fig. 19 for the atmospheric GROWIAN data are slightly lower compared to the modeled Kaimal data in Figs. 6 and 8. The higher complexity of real wind fields included wind events characterized by stronger differences of the wind speed over the y-z plane within the stretched wind fields, which are likely to explain such particular discrepancies. However, Figs. 17, 18 and 19 show a good agreement between the statistical properties and the DEL estimations between the original and the reconstructed signals of the CoWP, and the simulated bending moments from the atmospheric measured GROWIAN data. These findings are in agreement with the results shown in Sect. 4.1 for the modeled

365

370

standard Kaimal data. Therefore, it was demonstrated that the description of the dynamics provided by the coefficients  $D^{(1,2)}$  from the CoWP can be used as parameters for modeling the low-frequency signals of the tilt and yaw bending moments at the main shaft of a WT.

## 5 Conclusions and Outlook

375 The comparison between the low-frequency content of the center of wind pressure (CoWP) as a feature of a turbulent wind field and the low-frequency content of the BEM-simulated bending moments at the main shaft of a wind turbine, e.g., yaw and tilt, is performed. A strong correlation between these large-scale structures of the CoWP and bending moments has been quantified in terms of statistical properties, correlation factors, and damage equivalent load (DEL). This correlation is consistent with the results reported in the studies by Schubert et al. (2025) and Moreno et al. (2024), and it has been shown to be valid for wind fields from both atmospheric measurements and standard models. As a consequence of this correlation, a comprehensive description of the CoWP from a particular wind field (e.g., site-specific) might serve as a surrogate estimator of the low-frequency load events of the tilt and yaw bending moments at the main shaft of an operating wind turbine.

A step further is the utilization of a comprehensive understanding of the dynamics of the CoWP from wind data, with the objective of modeling loads. The stochastic Langevin approach has been proposed as a method for characterizing the dynamics of the CoWP. More interestingly, the method has been reverse-applied for the stochastic reconstruction of synthetic signals. The resulting statistics from the reconstructed signals and their estimated DEL have been shown to be comparable to those of the original CoWP and, more significantly, to those from the BEM-simulated bending moments. Consequently, the stochastic Langevin approach applied to the CoWP has been proven as a surrogate method for estimating the low-frequency content of the moments at the main shaft. In particular, the Langevin approach significantly reduces the computational cost by solving only a one- or two-dimensional stochastic equation instead of calculating a wind field at many different spatial points and its interaction with the turbine model. This has the potential for the reconstruction of very long modeled load data. This feature is essential for the assessment of the tilt and yaw bending moments when particularly large amounts of simulated data are required, e.g., for 25-year lifetime predictions under multiple wind conditions, and the computational costs associated with costly BEM simulations would thus be significantly reduced.

However, the development of lifetime predictions in engineering applications necessitates the incorporation of additional elements in conjunction with the proposed stochastic method for modeling the low-frequency component of the loads. Initially, a turbine-specific transfer function for rescaling the CoWP to the magnitudes of the low-frequency component of the bending moments should be derived. Secondly, a numerical model of the high-frequency component of the loads is required. A stochastic Gaussian model has been demonstrated to be a viable approach. Thirdly, site-specific wind characteristics should be considered. These characteristics should include the long-term standard wind conditions, such as the annual Weibull distribution of the wind speed. Additionally, spatial descriptions (i.e., perpendicular to the main flow) of the wind structures are necessary to describe the dynamics of the CoWP at the given location. These spatial descriptions may be derived either from

measured data over a two-dimensional area (e.g., using LiDAR techniques), or from accurately modeled wind data, which includes realistic information about the wind structures in the spatial domain. Once the three complementary elements have been resolved, the complete prediction of the yaw and tilt bending moments at the main shaft of a turbine can be applied as follows: site-specific wind data over relatively short intervals (e.g., 10 minutes), which are used for the calculation of the CoWP. Subsequently, the dynamics of the large-scale wind structures described by the CoWP are derived by using the Langevin stochastic approach. The parameters of the Langevin model for the specific wind conditions (i.e., drift and diffusion coefficients) are then estimated. Next, stochastic realizations of the low-frequency component of the loads are generated by combining the dynamics of the CoWP and the previously determined turbine-specific transfer function. Afterwards, the high-frequency component is modeled. Subsequently, the high- and low-frequency load signals, which have been modeled independently, are combined. Finally, the long-term distribution of the mean wind speed  $p(\bar{u})$  at the specific location is used to assess the entire lifetime damage of the bending moments (i.e., by applying the standard IEC procedure for load assessment based on mean wind speed binning and design load cases).

In the context of improved descriptions of the atmospheric turbulent wind, including the statistical and dynamical properties of the CoWP from atmospheric measured data into the standard wind models, could prove to be of significant value. Since the wind industry currently uses standard wind models for design and certification processes, the incorporation of atmospheric information would enhance the understanding of the aerodynamic interactions and more accurate load assessment of the turbines. For instance, wind structures such as gusts are assumed by the standard wind models to be homogeneous in space.

The CoWP has the capacity to grasp localized wind structures over the rotor plane. A parametrization of the dynamics of the CoWP from atmospheric wind would thus describe the realistic, likely non-homogeneous, spatial characteristics of the gusts. A comparison of the drift and diffusion coefficients derived from standard wind model data and measured data reveals that different characteristics of the wind fields are mapped into the Langevin equations. Consequently, the availability of local wind data enables the estimation of site-specific wind characteristics and the subsequent development of the stochastic load model.

This paper shows that the CoWP and its stochastic modeling represent a promising new tool for estimating the large-scale dynamics of specific loads at the wind turbine. The validity of this load estimation has been demonstrated in the context of the DELs. The dynamic response of modern wind turbines with increased size gives additional relevance to wind structures over the rotor plane. Larger areas covered by the increased rotors likely include inhomogeneities (e.g., severe differences in wind speed) over the rotor. In this direction, the CoWP and the stochastic approach delineated in our paper have the potential to serve as a tool for describing and modeling IEC extreme scenarios (i.e., with 50-year or 1-year return period). Up to now, calibrating the magnitude of the CoWP to the loads requires BEM simulations, at least on a finite time window. The validity of the CoWP approach to other loads at different turbine components remains to be investigated. For a single blade, a rotational frame of reference could be helpful for the calculation of the CoWP. Based on the results presented in this paper, it is recommended that a comparable procedure be considered for any other load in the turbine. The initial step involves normalizing the signals and validating the correlation. Following this, a stochastic model is to be configured to analyze and reconstruct the dynamic load response.

Data availability. The GROWIAN measurements, as well as the generated Kaimal wind fields can be obtained upon request.

# Appendix A: Correlation between DEL and DEL<sub>CoWP</sub>

The DEL<sub>CoWP</sub> introduced in Eq. (12) is based on the conclusion stated by Schubert et al. (2025) that large amplitude load events, lasting longer than 10s, e.g., bumps structures, drive large values of the DEL when using the Wöhler exponent m = 10. Now we present a different proof of this finding.

The aim is to compare the DEL between time series, with and without, particularly large amplitude load events. Fig. A1(a) shows an excerpt of  $15 \times 10^3$  s containing the results of the DEL from the time series of the yaw moment  $T_{\text{yaw}}$ . The horizontal red bars depict the period T = 10 min over which the DEL is calculated. The largest DEL are identified and visually separated above the horizontal gray line. The respective time series  $T_{\text{yaw}}$  from which the DEL are calculated are shown in (b). The darker highlighted load events at 3300, 3700, 3850, and 9800s correspond to the largest DEL (over the gray line) in (a). The zoom plot in the lower part of (b) illustrates the load event at  $t \approx 9800$ s.

Figure A1. Damage Equivalent Loads (DEL) of the yaw moment signal  $T_{\text{yaw}}$  from BEM simulations. (a) DELs of the  $T_{\text{yaw}}$ . The horizontal gray line at DEL = 1.1 visually separates the few largest DELs. The DELs are calculated with m = 10 over periods T = 10 min. An overlapping period of 5 min is considered between two consecutive intervals. The length of the horizontal red bars depicts the periods  $T_{\text{yaw}}$  with highlighted large events, which are inducing the largest DEL in (a). The length of such identified load events within the time series  $T_{\text{yaw}}$  is considered as 20 s over which the peak amplitude is included. The event at  $t \approx 9800$  s is detailed in the zoomed plot. The time series  $T_{\text{yaw}}$  are calculated by BEM simulations of the 5MW NREL turbine (see Sect. 3). The time series  $T_{\text{yaw}}$  are low-pass filtered with cutoff t = 0.1 and normalized to zero mean and standard deviation equal to 1.

Fig. A2(a) shows a modified time series ' $T_{yaw}$ -Mod' from which the large load events highlighted in Fig. A1(b) have been removed. The resulting DEL from  $T_{yaw}$ -Mod are shown in (b). The comparison between the DELs in Fig. A2(b) and Fig. A1(a), i.e., from the two versions of the time series  $T_{yaw}$ , confirms that large amplitude events in the signal induce large values of the DEL. Therefore, an accurate fatigue assessment of the moments T based on the DEL, as the standard procedure within the wind industry, requires an accurate description of such large amplitude loading events. The DEL<sub>CoWP</sub> is proposed in Sect. 4.1 as an approach for predicting those events on the loads from the wind field.

Figure A2. Damage Equivalent Loads (DEL) of a modified signal of the yaw moment ' $T_{\text{yaw}}$ -Mod' from BEM simulations. (a) Time series ' $T_{\text{yaw}}$ -Mod' from which the highlighted intervals in Fig. A1(b) have been removed (b) DELs of the  $T_{\text{yaw}}$ -Mod. The horizontal blue line at DEL = 1.1 is kept as a reference. The DELs are calculated with m = 10 over T = 10 min. An overlapping period of 5 min is considered between two consecutive intervals. The length of the horizontal red bars depicts the length of T.

## 455 Appendix B: DELs from low- and high-frequency components of the loads

460

The contribution of the low- and high-frequency components of the load signal to the DELs of the total load is investigated. Therefore, different components of the load signal are independently investigated. The low-frequency component ('Low freq.') corresponds to the low-pass filtered load described in Sec. 4, with a cutoff frequency of  $f_{cutoff} = 0.1$ Hz. The high-frequency component ('High freq.') corresponds to the load fluctuations with frequency over  $f_{cutoff}$ . The total load ('Total') is the estimated load from BEM simulations, which aggregates both the high- and the low-frequency components.

Figure B1(a) shows the DELs of the components of the load over an excerpt of 1200s. Each of the horizontal bars represents the period T = 60s over which the DEL is calculated. A fourth signal, 'Sum L+H', is included in the comparison. It corresponds to the combination of the DELs (i.e., not the time series) from the low- and high-frequency signals as,

$$(DEL_{Sum L+H}) = \alpha (DEL_{Low}) + \beta (DEL_{High}), \tag{B1}$$

calculated for each period T. The parameters  $\alpha$  and  $\beta$  are fitting parameters to achieve  $DEL_{Sum\ L+H} \approx DEL_{Total}$ . These parameters depends on the Wöhler coefficient m and the length T for the calculation of  $DEL_{Low}$  and  $DEL_{High}$ . In Fig. B1(a), the

parameters are  $\alpha = 1.2$ , and  $\beta = 0.5$ . The values of the coefficients  $\alpha$  and  $\beta$  from the load signals can be taken as weighting factors, indicating a dominating contribution of the DEL<sub>Low</sub> with respect to DEL<sub>High</sub>. For the case shown, the proportion is approximately 2:1.

Fig. B1(b) shows the correlation between the DELs of the Total, and the Sum L+H signals. The correlation is calculated for the DELs along the entire data set (4.7x10<sup>4</sup> s). Based on the strong correlation between the DELs in (b), the DELs of the total load might be interpreted as a weighted sum of the individual DELs from the low- and high-frequency components.

Figure B1. (a) 20-min excerpt of the DELs for the  $T_{titlt}$  signals at the main shaft. The length of the individual horizontal bars depicts the periods T. (b) Correlation plot between the total load and the sum of the DELs from the low- and high-frequency signals. The DELs are calculated with m = 10 over periods of T = 60s with an overlap of 30s between periods. The load signals are those calculated from BEM simulations of the 5MW NREL WT with Kaimal fields, with  $\bar{u} = 7$ m/s (see Sec. 3).

The results in Figs. B1 show that despite the dominance of the low-frequency contribution, both the low- and high-frequency components have important contributions to the DEL of the total load signal. Therefore, for a complete calculation of the DELs on the WT, a second model for the high-frequency contribution is required. The use of Gaussian distributed noise is proposed as a first approach. Three random Gaussian realizations, 'R1', 'R2', and 'R3', with the statistics from the original high-frequency load signal, are generated. The considered statistics include not only the mean and standard deviation, but also the correlation and dominant frequency. Next, the three Gaussian realizations of high-frequency fluctuations are added to the low-frequency component of the load. Then, the DELs are calculated.

Fig. B2(a) shows a 20-min excerpt of the DELs. For comparison, the original Total load, which is the sum of the original high-frequency and low-frequency components, is also compared. Fig. B2(b) shows a box-plot of the DELs over the entire time series  $(4.7 \times 10^4 \, \text{s})$ .

Figure B2. DELs of the total signals (low-frequency and high-frequency signals). (a) 20-min excerpt with individual DELs. (b) Box-plots of the DELs along the entire time series. The four high-frequency signals ('High freq.', 'R1', 'R2', and 'R3') have been added to the same low-frequency signal. The DELs are calculated with m = 10 over periods of T = 60s with an overlap of 30s between periods.

The comparability between the boxplots in Fig. B2 shows that Gaussian noise, with parametrized dominant frequency and correlation, can be used as a model for the high-frequency component of the load signals. Then, this Gaussian model for high-frequency fluctuations might be used in combination with our proposed model, based on the CoWP, which reproduced the low-frequency component of the load signal. Then, an entire description of the load signals could be achieved. The joint use of these two models must be further validated by comparing them to the total simulated loads. For that, a transfer function is required for scaling the magnitudes of the loads. The validation is out of the scope of this paper.

485

# **Appendix C: Correlation and structure function of the CoWP**

Fig. C1 shows the correlation function for the two components of the CoWP  $\langle CoWP_i(t) \ CoWP_j(t+\tau) \rangle$  for i=y,z. From top to bottom, the three rows show the correlation for  $i \neq j$ , i=j=y, and i=j=z, respectively. The panels on the left [(a),(c),(d)] correspond to the modeled Kaimal data. The panels on the right [(b),(d),(e)] show the atmospheric measured GROWIAN data. The correlation is calculated for time lags  $\tau = [-200\ 200]$ .

**Figure C1.** correlation function for the two components of the CoWP  $\langle CoWP_i(t) \ CoWP_j(t+\tau) \rangle$  for i=y,z and  $\tau=[-200\ 200]$ . In (a),(c),(e) for the modeled Kaimal data and in (b),(d),(f) for the atmospheric GROWIAN data.

## Appendix D: 'Special events' of the CoWP from GROWIAN data

In Fig. 14, particular loops are observed when correlating the CoWP and the tilt and yaw bending moments from atmospheric GROWIAN wind fields. Fig. D1 shows three exemplary temporal sequences corresponding to some of the observed loops in the correlation plots. Short intervals of 4s are shown. Panels (a) and (b) show sequences observed in the correlation between  $CoWP_y$  and  $T_{yaw}$  in Fig. 14(a). Panel (c) shows a sequence observed in the correlation between  $CoWP_z$  and  $T_{tilt}$  in Fig. 14(b). As observed in the three sequences, very strong differences of the wind speed over the rotor plane. The green dot shows the CoWP. The red dot shows a scaled version of the CoWP, which allows for better visualization. The scaling is done by subtracting a mean wind speed from all the points of the wind field. This subtraction is analogous to removing the mass of the beam when calculating the *center of mass* induced by external masses. In that way, larger distances of the CoWP with respect to the reference point are obtained.

Figure D1. Temporal evolution of the GROWIAN wind field and the CoWP on the y-z plane. The wind speed u(y,z,t) is color-coded. The green dot depicts the location of CoWP. For better visualization, the red dot depicts the location of a scaled version of the CoWP. The lines show the center lines of the grid, i.e., the reference location for calculating CoWP.

The relatively large deviations of the CoWP depicted by the loops in Fig. 14 from the GROWIAN data suggest that the CoWP might be very sensitive to such extreme differences of the wind field over the *y-z* plane at a given time step.

# Appendix E: Characteristics of the CoWP from standard Kaimal wind fields with shear

The results of the drift and diffusion coefficients  $D^{(1,2)}$  from the stochastic Langevin method applied to the CoWP from the GROWIAN measurements are shown in Fig. 15. To investigate and compare the effect of the shear in the dynamics of the CoWP from standard modeled wind fields, we calculated the coefficients  $D^{(1,2)}$  from Kaimal wind fields with a shear exponent of 0.2. The results are shown in Fig. E1. Interestingly, the superimposition of shear to the Kaimal wind fields results in additive noise only shifted towards higher heights.

515

**Figure E1.** Langevin approach of the CoWP from Kaimal wind fields with shear exponent of 0.2. (a) Drift coefficient  $D^{(1)}$ , and (b) Diffusion coefficient  $D^{(2)}$ . In black for the vertical component CoWP<sub>u</sub>, and in red for the horizontal component CoWP<sub>z</sub>.

Additionally, the trajectories of the CoWP on the y-z plane are shown in Fig. E2 for the original CoWP and a reconstructed signal CoWP $_R$  from the Kaimal wind fields with shear. In comparison to the trajectories from the atmospheric GROWIAN data in Fig. 16, the trajectories of the CoWP from the shear Kaimal wind fields are symmetric in the y-z directions. Again, only a vertical shift is observed within the CoWP, including shear effects, compared to Fig. 10.

Figure E2. Trajectories on the y-z plane of (a) the CoWP from the original shear Kaimal wind fields, and (b) the corresponding stochastically reconstructed signal CoWP $_R$ . The data are not normalized.

Author contributions. DM: Simulations, data analysis and calculations. JF, CS and MW: Review, analysis, discussion of the results, and contributions to the text. GR: Discussion of the results JS and GP: Discussion on the results from the manufacturer/operator perspective. JP: Extensive understanding of the method, analysis of the results, supervision, reviewing, and editing the text.

Competing interests. The authors declare no conflict of interest. An author is a member of the editorial board of journal WES.

Acknowledgements. We gratefully appreciate the valuable discussions with our partners, the Institute for Mechanical and Industrial Engineering Chemnitz and Nordex Energy SE, involved in PASTA project (Precise design methods of complex coupled vibration systems of modern wind turbines in turbulent conditions). The aim of this work was initiated as a hypothesis for the challenges discussed within the project.

This work has received funding from the German Federal Ministry for Economic Affairs and Climate Action according to a resolution by the German Federal Parliament (PASTA, 03EE2024).

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
