# Peer review of "From the center of wind pressure to loads on the wind turbine: A stochastic approach for the reconstruction of load signals"

_Wind Energy Science, 2025_

## Referee Comment (RC2)

**Review of WES 2025-78**

This article builds on the recently introduced concept of the center of wind pressure (CoWP) in turbulent inflow fields and shows a stochastic surrogate model for the position in time of CoWP. Good statistical agreement is shown between the reconstructed CoWP and BEM-estimated yaw bearing bending moments. Aerodynamic imbalances have been long linked to yaw bearing and rotor shaft bending moments. However a systematic method such as the introduction of the CoWP is novel. Regarding the extension of this concept and the development of the load surrogate model proposed in this article, some things to note below.

The authors state that the surrogate model is very fast, and can generate long timeseries. Thus it can be used to replace uncertain load extrapolation techniques or replace costly long-term simulations of the wind turbine – at least for certain load components. This is a very interesting prospect, but it's not thoroughly demonstrated in the paper. Regarding long-term load extrapolation (for example – loads with a one- or fifty-year occurrence probability) the PDF plots shown in Fig. 11 (and many other throughout the paper) – despite showing good agreement even in the tails of the PDFs, only reach relatively high levels of probability. I would recommend to show the ability of the method to predict extreme loads with a one-year occurrence period – which should not e computationally too intensive to achieve with a "traditional" BEM simulation-based approach. In alternative, authors could try to compare the proposed surrogate to existing long-term datasets in the literature. The dataset generated by Barone et al. – also used by Dimitrov and Zhang (cited in manuscript) in their study – contains long-term extreme loads for the same testcase used in this manuscript. Alternatively, the dataset by Papi and Bianchini contains 50 years of loads for the NREL 5MW – albeit on a floating foundation. Please note that other references may exist, although I am not aware of them. Here are the mentioned references (https://www.sandia.gov/app/uploads/sites/273/2025/02/AIAA2012-1288-SAND2011-3780C.pdf https://www.osti.gov/biblio/1078621 https://zenodo.org/records/10514143 )
As per Journal reviewer guidelines, feel free to use or not use them as you see fit.

Some aspects of the introduction could be clarified. In particular:
L26: "However, they do not yet incorporate turbulent flow structures." – Spectral models include spatial coherence functions. They do not explicitly resolve eddies; I imagine this is what authors intend here. Please clarify.

L27-30: Why are increased dimensions related to additional uncertainty in the impact of turbulent inflow on loads?

L35-40: Can more details be added regarding the observation of manufacturers: "According to manufacturers and operators of WTs, numerical simulations of the specific WTs and the standard IEC wind modeling assumptions do not adequately reflect certain load events that may be important for the structural integrity of the machines in operation."

L45-53: This paragraph appears a bit confused. Some works on numerical models are mixed with works on load extrapolation techniques and work on control techniques. Please reorganize this section in the context of the introduction.

L190: is data also filtered for direction? If the flowfield is misaligned with respect to the inflow how may this affect the measured coherence of the eddies and the results in this study?

The way the GROWIAN data is stretched is unclear. Is it a mix of interpolation and extrapolation? More details would be requited here. Moreover, is wind direction included in the dataset? Wouldn't changes in the man incoming wind field affect the measured coherence and size of the eddies?

Results: The BEM results are low-pass filtered as CoWP is a good description of large-scale turbulent fluctuations. The signals are also zero-meaned and normalized to have a standard deviation of 1. In the context of developing a surrogate model the manipulations that are done to the data seem to be significant. What is the effect on the long-term statistics and extrapolated loads of the filtered-out high-frequency component?

Results: Regarding the normalization of the signals – given the excellent statistical agreement between the normalized signal statistics, it would be interesting to see a transfer function mapping the CoWP to yaw bearing bending moments or other wind turbine load sensors as the author see fit.

Figure 8: When commenting this figure I would highlight the fact that the DELS agree well in an aggregate sense, but less so on a simulation per simulation perspective. Indeed, while statistics are in very good agreement (c, d) and correlation is good (a, b) a large spread in the data con be seen in figures 8 (a) and 8 (b).

Finally, please provide more details on the BEM numerical setup. Some details are included in the provided reference but should be repeated herein since the simulations constitute the reference for the entire work.

---

## Author Comment (AC1)

**Response to Referee 1**

**From the center of wind pressure to loads on the wind turbine: A stochastic approach for the reconstruction of load signals**

Referee's comment (RC) in blue
Author's comment (AC) in black

The references to lines in the manuscript (e.g., 'L.80') are given with respect to the **new version** of the paper.
In gray: text from the revised version of the manuscript.

**GENERAL COMMENTS**
REFEREE:
The method proposed by the authors is intended to enable a fast estimation of the Damage Equivalent Load (DEL) over very long time histories (years). The comparison between DEL, $\mathrm{DEL}_{CoWP}$, and $\mathrm{DEL}_{CoWP_R}$ is satisfactory, but not excellent: the slope of the interpolation line is clearly below one, and there is a notable spread of data around the fitted line. The authors do not comment on how this discrepancy might affect the DEL estimation over long time periods. Should we expect differences in the order of 1%, 5%, 10%, 30%, or even 50%?

Thank you for your comment. We recognize that the current capacity of the method was not accurately stated in the manuscript. At its current state, our method is not sufficient for calculating the DELs over long-time periods. Instead, we present simple concepts with which the calculation of the long-period DELs will be easily possible.

Further considerations should be considered with the method proposed in the paper, for the complete calculation of the DELs over the lifetime of a WT. First, a transfer function for calibrating the normalized magnitudes of the CoWP to the magnitudes of the bending moments should be derived, for example, from corresponding data of a wind turbine of interest. Second, a model for the high-frequency component of the loads needs to be implemented. Third, the long-term wind conditions (e.g., Weibull distribution) at a specific location must be known.

A few lines have been incorporated and/or modified in the manuscript to correctly describe the current capability of the method, as well as to state the future steps to be included for a complete assessment of the loads in the context of long-time calculations.

L.82: In its current state, the method is limited to the modeling of the dynamics of the low-frequency components of the bending moments. However, when combined with a description of the high-frequency components, a validated rescaling procedure, and the characterization of the site-specific wind conditions, this approach enables a novel method for a fast assessment of the lifetime loads in WTs.

L.166: However, the implementation of the stochastic method for lifetime load assessment in engineering applications (i.e., including both the high- and low-frequency components of the loads) is constrained in its application. To ensure a comprehensive lifetime model, it is necessary to incorporate additional elements. A turbine-specific transfer function for rescaling the magnitudes of the loads is required. A complementary model for the high-frequency components of the loads must be integrated. Finally, the occurrence of the loads must be weighted by the distribution of the mean wind speed at the location of the WT (e.g., annual Weibull distribution).

The method can predict DELs associated with the low-frequency component of the loads. However, the overall DEL also includes high-frequency loads, which are characterized by many cycles. The authors do not comment on the difference between DELs from low-frequency load components and total DELs.

Thank you for the comment. In fact, the difference between the DELs from low-frequency load components and DELs from total loads has not been discussed in the paper. In the following, an analysis of the contribution to the DELs of low-frequency and high-frequency components of the loads is shown.

Three signals are compared. The low-frequency signal ('Low freq.') corresponds to the low-pass filtered load described in Sec. 4 in the manuscript with a cutoff frequency of $f_{cutoff} = 0.1$Hz. The high-frequency signal ('High freq.') corresponds to the load fluctuations with frequency over $f_{cutoff}$. The total load ('Total') is the estimated load from BEM simulations, which aggregates both the high- and the low-frequency contributions. The investigated load signals are those calculated for the 5MW NREL WT with Kaimal fields, with $\bar{u} = 7$m/s (see Sec. 3 in the manuscript).

Figure A shows an excerpt of the time series of the three signals (Total, Low freq., and High freq.) over 1200s for the $T_{titlt}$ at the main shaft of the WT.

[Figure]

Figure A: 20-min excerpt of the load signals (Total, Low freq., and High freq.), for the $T_{titlt}$ at the main shaft.

Now, the DELs of the three signals are investigated. Fig. B(a) shows the DELs calculated over periods T = 60s along the same 1200s shown in Fig. A. A fourth signal, 'Sum L+H' is included. It corresponds to the combination of the DELs (i.e., not the time series) from the low- and high-frequency signals as,

$$(\mathrm{DEL_{Sum\ L+H}}) = \alpha\,(\mathrm{DEL_{Low}}) \,+\, \beta\,(\mathrm{DEL_{High}}), \tag{1}$$

calculated for each period T. The parameters $\alpha$ and $\beta$ are fitting parameters to achieve $\mathrm{DEL_{Sum\ L+H}} \approx \mathrm{DEL_{Total}}$. These parameters depends on the Wöhler coefficient $m$ and the length T for the calculation of $\mathrm{DEL_{Low}}$ and $\mathrm{DEL_{High}}$. In Fig. B(a), the parameters are $\alpha = 1.2$, and $\beta = 0.5$. The values of the coefficients $\alpha$ and $\beta$ from the load signals can be taken as weighting factors, indicating a dominating contribution of the $\mathrm{DEL_{Low}}$ with respect to $\mathrm{DEL_{High}}$. For the case shown, the proportion is approximately $2:1$.

Fig. B(b) shows the correlation between the DELs of the original 'Total', and the added 'Sum L+H'. The correlation is calculated for the DELs along the entire data set $(4.7\mathrm{x}10^4\,\mathrm{s})$.

[Figure]

Figure B: (a) 20-min excerpt of the DELs for the $T_{titlt}$ signals at the main shaft. The length of the individual horizontal bars depicts the periods T. (b) Correlation plot between the total load and the sum of the DELs from the low- and high-frequency signals. The DELs are calculated with $m = 10$ over periods of T = 60s with an overlap of 30s between periods.

In our paper, we concentrated entirely on the low-frequency events. This choice is based on the particular interest of our research partners. However, for a complete calculation of the DELs on the WT, a second model for the high-frequency component is required. The use of Gaussian distributed noise is proposed as a first approach. Three random Gaussian realizations, 'R1', 'R2', and 'R3', with the statistics from the original high-frequency load signal, are generated. The considered statistics include not only the mean and standard deviation, but also the correlation and dominant frequency. Fig. C(a) shows an excerpt of the time series of the original and the random Gaussian signals. In (b), the corresponding PDFs are shown. A Gaussian distribution fitted to the PDF of the original 'High freq.' is depicted by the solid black line.

[Figure]

Figure C: High-frequency signals of the original load signal ('High-freq') and three realizations of Gaussian-distributed noise ('R1', 'R2', and 'R3'). (a) 20-min excerpt of the time series. (b) PDFs of the signals. The solid line in (b) depicts a Gaussian distribution.

Next, the four high-frequency signals are added to the low-frequency component of the load. Then, the DELs are calculated. Fig. D(a) shows a 20-min excerpt of the DELs. Fig. D(b) shows a box-plot of the DELs over the entire time series $(4.7\text{x}10^4\,\text{s})$.

[Figure]

Figure D: DELs of the total signals (low-frequency and high-frequency signals). (a) 20-min excerpt with individual DELs. (b) Box-plots of the DELs along the entire time series. The four high-frequency signals ('High freq.', 'R1', 'R2', and 'R3') have been added to the same low-frequency signal. The DELs are calculated with $m = 10$ over periods of T = 60s with an overlap of 30s between periods.

The comparability of the PDFs in Fig. C and the boxplots in Fig. D shows that Gaussian noise, with parametrized dominant frequency and correlation, can be used as a model for the high-frequency component of the load signals. Then, this Gaussian model for high-frequency fluctuations might be used in combination with our proposed model, based on the CoWP, which reproduced the low-frequency component of the load signal. Then, an entire model of the load could be achieved. The joint use of these two models must be further validated by comparing them to the total simulated loads. For that, a transfer function is required for scaling the magnitudes of the loads.

Given the relevance of commenting about the DEL calculated from the low-frequency component and the total load signals, **Appendix B** has been incorporated into the manuscript, which partially includes the analysis previously

shown. Additionally, the lines below have been modified.

L.293-299: It is essential to acknowledge that the discussion on the DELs presented in our work is exclusively focused on the DELs from the low-frequency component of the signals. This choice is based on a particular interest of our research partners. In order to assess the complete DELs (e.g., from both the low- and high-frequency load events), it is necessary to establish an additional model for incorporating the contribution from the high-frequency signal. In this direction, a simple surrogate stochastic model has shown satisfactory results. The characteristics of the original high-frequency load signal are well reproduced. The proposed stochastic model for the high-frequency signals, and calculations on the differences between the DELs from the low- and high-frequency load components, and total DELs are shown in Appendix B.

The simulations using Kaimal data are performed at a single mean wind speed, under partial load conditions (region II). What is the impact of turbine control on the accuracy of the proposed method? It would be useful to compare the results with wind speeds above the rated.

Thank you for the comment. You are right, in the paper, only a single mean wind speed, included within the region II, is presented. In the following, we show the results of the analysis for the case with $\bar{u} = 13$m/s (i.e., above the rated), by using standard Kaimal wind fields.

Figure E shows the time series of $\bar{u}$ at hub height. Fig. F is equivalent to Fig. 5 in the manuscript. It shows the comparison between the time series of the normalized signals of the CoWP and the bending moments at the main shaft.

[Figure]

Figure E: 20-min excerpt of the wind speed $u(t)$ at hub height.

[Figure]

(a)

(b)

Figure F: 20-min excerpts of the CoWP and the bending moments at the main shaft of a WT. (a) $T_{\text{yaw}}$ and $\text{CoWP}_y$, and (b) $T_{\text{tilt}}$ and $\text{CoWP}_z$. The signals are normalized and low-pass filtered.

Figure G is equivalent to Fig. 6 in the manuscript. The correlation between the signals is shown.

[Figure]

Figure G: CoWP against the bending moments plotted as $(CoWP(t), T(t))$ for each time step $t$ of the time series. In (a) $T_{\text{yaw}}$ and $\text{CoWP}_y$, and (b) $T_{\text{tilt}}$ and $\text{CoWP}_z$. The gray lines depict linear fittings $T = a\,(CoWP) + b$. The signals are normalized and low-pass filtered.

Lastly, Fig. H is equivalent to Fig. 8 in the manuscript. The correlation plots between the DEL and $\text{DEL}_{\text{CoWP}}$ are shown in (a) and (b). The corresponding box plots are presented in (c) and (d).

[Figure]

Figure H: Comparison between the DEL and $DEL_{CoWP}$. Correlation plots and box plots for $CoWP_y$ and $T_{yaw}$ in (a) and (c), and $CoWP_z$ and $T_{tilt}$ in (b) and (d). The gray lines in (a) and (b) depict linear fittings. In the box plots in (c) and (d), the horizontal line inside each box shows the median, and the bottom and top edges indicate the 25th and 75th percentiles. The whiskers indicate the most extreme data points. The markers show outliers. The DEL and $DEL_{CoWP}$ are calculated with $m = 10$ over periods $T = 60\,s$ with $30\,s$ overlapping between two consecutive periods. The signals are normalized and low-pass filtered.

Both, the correlation between the time series of the CoWP and the bending moments (Figs. F and G), and their corresponding DEL and $DEL_{CoWP}$ in Fig. H, show that the results for $\bar{u} = 13$m/s are comparable to those presented in the paper for $\bar{u} = 7$m/s. Accordingly, the proposed method for estimating large-scale dynamics of the bending moments based on the CoWP from the wind holds for wind speeds above rated.

Accordingly, L.291 has been added to the manuscript.

The validity of the method has been proven for the rated power regime of the WT.

There is no discussion on how the proposed method could be integrated into current engineering practice, which involves estimating total DELs (including both low- and high-frequency components) from 10-minute simulations. In other words, how can the DELs associated with low-frequency atmospheric variations (predictable using the CoWP or its surrogate) be integrated with those due to high-frequency atmospheric fluctuations?

Thank you very much for your comment. The authors agreed on the high relevance of the question. Consequently, we included a paragraph in the conclusions of the manuscript L.393-412 for discussing how the proposed method can be incorporated into engineering applications.

However, the development of lifetime predictions in engineering applications necessitates the incorporation of additional elements in conjunction with the proposed stochastic method for modeling the low-frequency component of the loads. Initially, a turbine-specific transfer function for rescaling the CoWP to the magnitudes of the low-frequency component of the bending moments should be derived. Secondly, a numerical model of the high-frequency component of the loads is required. A stochastic Gaussian model has been demonstrated to be a viable approach. Thirdly, site-specific wind characteristics should be considered. These characteristics should include the long-term standard wind conditions, such as the annual Weibull distribution of the wind speed. Additionally, spatial descriptions (i.e., perpendicular to the main flow) of the wind structures are necessary to describe the dynamics of the CoWP at the given location. These spatial descriptions may be derived either from measured data over a two-dimensional area (e.g., using LiDAR techniques), or from accurately modeled wind data, which includes realistic information about the wind structures in the spatial domain. Once the three complementary elements have been resolved, the complete prediction of the yaw and tilt bending moments at the main shaft of a turbine can be applied as follows: site-specific wind data over relatively short intervals (e.g., 10 minutes), which are used for the calculation of the CoWP. Subsequently, the dynamics of the large-scale wind structures described by the CoWP are derived by using the Langevin stochastic approach. The parameters of the Langevin model for the specific wind conditions (i.e., drift and diffusion coefficients) are then estimated. Next, stochastic realizations

of the low-frequency component of the loads are generated by combining the dynamics of the CoWP and the previously determined turbine-specific transfer function. Afterwards, the high-frequency component is modeled. Subsequently, the high- and low-frequency load signals, which have been modeled independently, are combined. Finally, the long-term distribution of the mean wind speed $p(\bar{u})$ at the specific location is used to assess the entire lifetime damage of the bending moments (i.e., by applying the standard IEC procedure for load assessment based on mean wind speed binning and design load cases).

The analysis focuses on DELs. However, wind turbines are also designed to withstand ultimate loads. Could the proposed approach be useful in this context as well? Is there any correlation between CoWP and ultimate hub loads?

Thank you very much for the question. As you have appropriately mentioned, the assessment of ultimate loads is as essential as the corresponding fatigue predictions (e.g., DELs). In principle, the CoWP might be directly correlated to ultimate loads on the WT. When coherent structures, such as localized gusts, are responsible for the ultimate loads on the WT, our CoWP approach could correctly predict the maximum loads.

However, the correlation between extreme loads and localized structures remains an open question for the wind industry. The gusts defined by the IEC under extreme conditions are assumed to be homogeneous (i.e., over the entire rotor plane). For modern large wind turbines, however, it is expected that gusts are spatially inhomogeneous and affect the rotor as localized structures. Further analysis of atmospheric wind fields is needed. Long-term wind field measurements are required. Accurate modeling of localized, coherent wind structures would enable our approach to be applied to ultimate load calculations.

The section on data availability is missing. If possible, the authors should share both the data and the scripts used.

Thank you for the suggestion. We are working on providing the scripts and an exemplary data set. Together, they could be used for calculating the CoWP and applying the stochastic method for reconstructing random signals of the loads.

**SPECIFIC COMMENTS**
**Section 2.2 (Damage Equivalent Load):** This section summarizes well-known information about DEL calculation, already available in IEC guidelines. It should be omitted.

We appreciate your recommendation. However, we consider that describing the DEL along Section 2.2 and in Eq.(5) in the manuscript is essential for the afterward introduction of the $\text{DEL}_{\text{CoWP}}$ in Eq.(12).

Nevertheless, the section has been shortened by removing some details that, as you mentioned, are not highly relevant for the manuscript and are already available in the IEC standard.

**Figure 8:** What exactly do the whiskers represent? The caption mentions they indicate the most extreme data points, but does not specify how these are defined (e.g., 95% or 99% confidence interval?).

The whiskers in the boxplots (Figs. 8a, 8b, 13a, 13b, 19a and 19b) represent the outlier threshold based on the interquartile range (IQR) method. In this case, the $1.5 \times$ IQR rule is applied. Then, the length of the whiskers are calculated as $Q1 - (1.5 \times \text{IQR})$ and $Q3 + (1.5 \times \text{IQR})$, with $\text{IQR} = Q3 - Q1$.

A more complete definition of the whiskers is provided in the caption of the figures. Note that instead of quartiles, percentiles are used in the paper.

**Lines [prev.]221–222:** The cut-off frequency used for filtering CoWP and loads should be close to the 3P frequency. However, the authors use 0.1 Hz, which is significantly lower than the 3P frequency of the NREL 5MW turbine operating at rated power (approximately 0.6 Hz). The reason for this choice is unclear. The type and order of the filter should also be specified.

Thank you for noticing this inconsistency.

The mean wind speed at the hub is set to $\bar{u} = 7\text{m/s}$ (see Sect. 3). Then, the NREL 5MW is operating at around 9rpm, or P $\approx 0.15$Hz. Correspondingly, $3\text{P} \approx 0.45$Hz. However, it is not true that the applied filter is of the order of the 3P frequency, as we mistakenly stated in L.221.

Certainly, the cutoff frequency of the filter should be lower than the 3P and P frequencies to remove the effect of the gravitational loads. Therefore, the selection of $f_{cutoff}$ is rather constrained by the P frequency.

Fig. I shows the spectrum $E(f)$ of the CoWP in the horizontal direction and the yaw bending moment. As observed, the agreement on the frequency content between the signals remains up to $f = 0.1$Hz. At higher frequencies, the load signal deviates to larger fluctuations. Accordingly, only wind and load events with $f > 0.1$Hz (i.e., with temporal scales larger than 10s are investigated).

[Figure]

Figure I: Energy spectrum of the CoWP in the horizontal direction and the yaw bending moment.

The filter is a finite impulse response (FIR) filter with the pass-band frequency $f_{cutoff}$.

 has been modified in the manuscript.

The filter is a finite impulse response (FIR) filter with the cutoff or pass-band frequency $f_{cutoff}$. The value of $f_{cutoff}$ should be lower than the rotational frequency P of the WT.

**Figure 8:** Why is the correlation between DEL and $\text{DEL}_{CoWP}$ better for the $T_{yaw}$ loads (left-right CoWP displacement) than for $T_{tilt}$ loads (up-down CoWP displacement)?

Thank you for the question. A better agreement between the DEL and $\text{DEL}_{\text{CoWP}}$ for the $T_{yaw}$ compared to the $T_{tilt}$ in Fig.8 can be explained by the correlation between the time series of the CoWP and the bending moments shown in Fig. 6 in the manuscript.

In Fig. 6, the slopes indicate a very good correlation for both, $T_{yaw}$ in (a), and $T_{tilt}$ in (b), with respect to the CoWP. However, the results of the $T_{tilt}$ in (b) are slightly more scattered compared to $T_{yaw}$. The scattering is quantified by a higher value of the Root Mean Squared Error (RMSE). Such stronger or more frequent differences between the $T_{tilt}$ and the $CoWP_z$ explain the lower correlation of the corresponding DELs in Fig. 8(b).

We consider that your question is worth addressing in the manuscript.  has been added.

[In Fig.8] A lower correlation is obtained for the DEL and $\text{DEL}_{\text{CoWP}}$ in the vertical direction in panel (b) compared to the horizontal component shown in (a). The lower correlation is explained by the more scattered results within

the correlation of the time series of the $T_{\text{tilt}}$ and the $CoWP_z$ shown in Fig.6. There, a value of RSME $= 0.40$ indicates a higher degree of scattering for $T_{\text{tilt}}$, compared to a RSME $= 0.34$ for $T_{\text{yaw}}$.

**Figures 10, 16, [prev.]D2:** The CoWP is defined with respect to the center of the rotor disk. Therefore, the $CoWP_z$ should oscillate around 90 m (Kaimal data) or 125 m (GROWIAN data). This is not evident in the figures.

You are entirely right. The $CoWP_z$ should oscillate around 90 m for the Kaimal fields, and 125 m for the GROWIAN data. Accordingly, the trajectories of the CoWP are shown in Figs. 10, 16, and E2 exhibit the vertical component $CoWP_z$ (y-axis) around 90m or 125m, respectively. Notably, the trajectories in Fig. E2 are shifted upwards ($CoWP_z$ around 95 m) due to the wind shear.

**Figures 8a, 8b, 19a, 19b:** It would be helpful to include the RMSE values.

Thank you for the comment and suggestion.

The RMSE values have been included in all the figures illustrating a linear fitting (i.e., Figures 6a, 6b, 8a, 8b, 14a, 14b 19a, and 19b).

**A new version of the manuscript is provided along with a diff file**.

**References**

[1] Schubert, C., Moreno, D., Schwarte, J., Friedrich, J., Wächter, M., Pokriefke, G., Radons, G., and Peinke, J.: Introduction of the Virtual Center of Wind Pressure for correlating large-scale turbulent structures and wind turbine loads, Wind Energy Sci. Discuss. [preprint], 2025, 1–19, https://doi.org/10.5194/wes-2025-28, *in review*, 2025.

---

## Author Comment (AC2)

**Response to Referee 2**
**From the center of wind pressure to loads on the wind turbine: A stochastic approach for the reconstruction of load signals**

Referee's comment (RC) in blue
Author's comment (AC) in black

The references to lines in the manuscript (e.g., 'L.80') are given with respect to the **new version** of the paper.
In gray: text from the revised version of the manuscript.

**GENERAL COMMENTS**

REFEREE:
The authors state that the surrogate model is very fast, and can generate long time series. Thus it can be used to replace uncertain load extrapolation techniques or replace costly long-term simulations of the wind turbine – at least for certain load components. This is a very interesting prospect, but it's not thoroughly demonstrated in the paper. Regarding long-term load extrapolation (for example – loads with a one- or fifty-year occurrence probability) the PDF plots shown in Fig. 11 (and many other throughout the paper) – despite showing good agreement even in the tails of the PDFs, only reach relatively high levels of probability. I would recommend to show the ability of the method to predict extreme loads with a one-year occurrence period – which should not be computationally too intensive to achieve with a "traditional" BEM simulation-based approach.

In alternative, authors could try to compare the proposed surrogate to existing long-term datasets in the literature. The dataset generated by Barone et al. – also used by Dimitrov and Zhang (cited in manuscript) in their study – contains long-term extreme loads for the sametestcase used in this manuscript. Alternatively, the dataset by Papi and Bianchini contains 50 years of loads for the NREL 5MW – albeit on a floating foundation. Please note that other references may exist, although I am not aware of them. Here are the mentioned references ([...]). As per Journal reviewer guidelines, feel free to use or not use them as you see fit.

Thank you very much for your comment. We must admit that the main application of our current method was not stated accurately. The method could indeed be used to replace uncertain load extrapolation techniques or replace costly long-term simulations of the WT. However, additional steps are still required for achieving a complete assessment of the lifetime loads on a WT. First, transfer functions should be derived for rescaling the values of the CoWP to the actual low-frequency components of the loads. These transfer functions will depend on

the specific turbine characteristics. Second, an additional model must be added to reproduce the high-frequency component of the signals. Third, the incorporation of the statistics of local wind conditions (e.g., annual mean wind speed distribution) is required for calculating the long-time loads. Combined with these three numerical descriptions, our proposed low-frequency model, based on the CoWP, could replace costly long-term simulations for assessment of the loads on WTs.

Concerning the second additional element, namely an additional model to reproduce the high-frequency contribution of the load signals, we could evidence that a simple surrogate stochastic model can be used (We refer to this point later in this response letter. See 'RESULTS' section). Then, it remains open the turbine-specific transfer function and the site-specific wind conditions.

To partially address your comment and provide the correct scope of our model, a description of how the method could be applied for engineering applications (e.g., lifetime calculations including loads with fifty-year occurrence probability) has now been included as follows:

in L.82: In its current state, the method is limited to the modeling of the dynamics of the low-frequency components of the bending moments. However, when combined with a description of the high-frequency components, a validated rescaling procedure, and the characterization of the site-specific wind conditions, this approach enables a novel method for a fast assessment of the lifetime loads in WTs.

in L.166: However, the implementation of the stochastic method for lifetime load assessment in engineering applications (i.e., including both the high- and low-frequency components of the loads) is constrained in its application. To ensure a comprehensive lifetime model, it is necessary to incorporate additional elements. A turbine-specific transfer function for rescaling the magnitudes of the loads is required. A complementary model for the high-frequency components of the loads must be integrated. Finally, the occurrence of the loads must be weighted by the distribution of the mean wind speed at the location of the WT (e.g., annual Weibull distribution).

and, in the conclusion of the manuscript L.393-412,

However, the development of lifetime predictions in engineering applications necessitates the incorporation of additional elements in conjunction with the proposed stochastic method for modeling the low-frequency component of the loads. Initially, a turbine-specific transfer function for rescaling the CoWP to the magnitudes of the low-frequency component of the bending moments should be derived. Secondly, a numerical model of the high-frequency component of the loads is required. A stochastic Gaussian model has been demonstrated to be a viable approach. Thirdly, site-specific wind characteristics should be considered. These characteristics should include the long-term standard wind conditions, such as the annual Weibull distribution of the wind speed. Additionally, spatial descriptions (i.e., perpendicular to the main flow) of the wind

structures are necessary to describe the dynamics of the CoWP at the given location. These spatial descriptions may be derived either from measured data over a two-dimensional area (e.g., using LiDAR techniques), or from accurately modeled wind data, which includes realistic information about the wind structures in the spatial domain. Once the three complementary elements have been resolved, the complete prediction of the yaw and tilt bending moments at the main shaft of a turbine can be applied as follows: site-specific wind data over relatively short intervals (e.g., 10 minutes), which are used for the calculation of the CoWP. Subsequently, the dynamics of the large-scale wind structures described by the CoWP are derived by using the Langevin stochastic approach. The parameters of the Langevin model for the specific wind conditions (i.e., drift and diffusion coefficients) are then estimated. Next, stochastic realizations of the low-frequency component of the loads are generated by combining the dynamics of the CoWP and the previously determined turbine-specific transfer function. Afterwards, the high-frequency component is modeled. Subsequently, the high- and low-frequency load signals, which have been modeled independently, are combined. Finally, the long-term distribution of the mean wind speed $p(\bar{u})$ at the specific location is used to assess the entire lifetime damage of the bending moments (i.e., by applying the standard IEC procedure for load assessment based on mean wind speed binning and design load cases).

On the other hand, we appreciate the recommendation on using long-term data sets available in the literature. However, such data only comprises data on loads. The corresponding wind fields are required for calculating the CoWP and then validating our model.

As a side note, we would like to highlight that the application of the proposed stochastic approach is not limited to the estimation and extrapolation of loads. It also presents a method for modeling large-scale wind structures that the IEC standard wind models do not adequately describe. The IEC wind models assume the wind gusts to occur spatially homogeneously over the entire rotor plane. However, this assumption is not likely to be valid for modern wind turbines with rotor diameters around 200 m. Instead, gusts approach the rotors as localized structures. Thus, the question of the location of the gust becomes relevant. This location can be potentially grasped by the proposed CoWP.

Therefore, the second use of the method in the context of wind characterization should not be underestimated, as it proposes an extended description relatively simple but might be relevant for the structural life of the WT. In this regard, we reinforce in the manuscript the capability of the method for extended wind characterization.

L.78 in the introduction of the manuscript has been modified,

Our model thus offers a twofold approach. On the one hand, it facilitates the characterization and modeling of large-scale wind structures. The wind energy sector is in urgent need of a comprehensive description of these large-scale structures, as standard wind models are likely to oversimplify them. Modern

large wind turbines are particularly vulnerable to this oversimplification. On the other hand, our stochastic model [...]

L.416-419 in the conclusions have been added,

For instance, wind structures such as gusts are assumed by the standard wind models to be homogeneous in space. The CoWP has the capacity to grasp localized wind structures over the rotor plane. A parametrization of the dynamics of the CoWP from atmospheric wind would thus describe the realistic, likely non-homogeneous, spatial characteristics of the gusts.

**INTRODUCTION**

[prev.]L26: "However, they do not yet incorporate turbulent flow structures." – Spectral models include spatial coherence functions. They do not explicitly resolve eddies; I imagine this is what authors intend here. Please clarify.

Thank you for the comment and correction. The sentence L.29 has been modified in the manuscript.

However, they do not yet explicitly resolve the turbulent eddies, i.e., the spatial characteristics of the turbulent flow structures. For the spatial coherence of turbulence, an exponential decay with distance is assumed.

[prev.]L27-30: Why are increased dimensions related to additional uncertainty in the impact of turbulent inflow on loads?

Thank you for this question. The sentences have been reorganized to state clearly and emphasize that the additional uncertainty of the estimated loads is attributable to structural properties that are distinctive of modern larger turbines, i.e., higher flexibility of the blades.

The sentences, L.34-38, have been modified in the manuscript.

Recent advancements in WT design show a persistent trend towards increasing dimensions, including higher heights and larger rotor diameters. Accordingly, certain structural properties are significantly modified within the designs of the larger WTs. Specifically, a higher degree of flexibility is characteristic of the larger and slimmer rotor blades. This may raise concerns about the validity of the assumptions or the omission of specific turbulent structures within the aforementioned standard wind models currently used by the WT industry.

[prev.]L35-40: Can more details be added regarding the observation of manufacturers: "According to manufacturers and operators of WTs, numerical simulations of the specific WTs and the standard IEC wind modeling assumptions do not adequately reflect certain load events that may be important for the structural integrity of the machines in operation."

Thank you for the question. With this sentence, we want to mention that unexpected measurements drive the main motivation for our work on operating

WTs. However, further details on the measurements or details of the specific WTs are not allowed due to confidentiality reasons in the framework of our research collaboration with the manufacturer.

[prev.]L45-53: This paragraph appears a bit confused. Some works on numerical models are mixed with works on load extrapolation techniques and work on control techniques. Please reorganize this section in the context of the introduction.

Thank you for the comment. The referenced investigations have been introduced in a more comprehensive and structured way. The paragraph has been modified in the manuscript L.51-62.

A general requirement within the wind industry is to simplify the complexity of WT representations in turbulent wind environments to allow practical implementation and minimize computational costs. As stipulated in the standard guidelines (IEC, 2019), numerical simulations of a wide range of operational scenarios are required for the validation of WT designs. Consequently, optimizing the computational time and power is imperative while ensuring satisfactory accuracy of the estimations of the responses of the WT. Some approaches have been proposed to reduce the complexity of the interaction between the wind and the WT. Examples of approaches based on a given wind field include a modified actuator sector model (Mohammadi et al., 2024) and the calculation of extended equivalent wind speeds over the rotor area (Choukulkar et al., 2016). Conversely, techniques are employed to extract characteristics of the incoming flow field from load measurements at the WT, such as blade-load-based estimators (Coquelet et al., 2024). Furthermore, due to the limitations in computational power, the loads on the WT are typically estimated over short intervals, e.g., 10 minutes. Consequently, numerical techniques have been proposed for extrapolating the loads estimated from such short time scales to lifetime load scenarios containing extreme events (Zhang and Dimitrov, 2023; Qingshan et al., 2022).

[prev. ]L190: is data also filtered for direction? If the flowfield is misaligned with respect to the inflow how may this affect the measured coherence of the eddies and the results in this study?

Thank you for your question. Yes, the GROWIAN measurements have been filtered by direction to guarantee an undisturbed flow. A sentence, L.214, has been added to the manuscript to mention this information.

To guarantee an undisturbed flow, the wind direction over the 10-min periods remains within a 100° range with respect to the location of the two met masts (i.e., main direction of the flow).

The way the GROWIAN data is stretched is unclear. Is it a mix of interpolation and extrapolation? More details would be requited here. Moreover, is wind direction included in the dataset? Wouldn't changes in the man incoming wind field affect the measured coherence and size of the eddies?

Thank you for your comment. In fact, some details were missing. The details have been added to the manuscript in L.217-223.

The rescaled GROWIAN fields are defined on a stretched grid of $152\,\mathrm{m}\times150\,\mathrm{m}$. The stretching is performed by increasing the distance between neighboring points of the original grid by factors of 1.5 and 2 in the vertical and horizontal directions, respectively. The green circles in Fig.4 illustrate the rescaled GROWIAN spatial arrangement, centered at $y = 0\,\mathrm{m}$ and $z = 125\,\mathrm{m}$. The wind speed measurements at the 16 original locations have not been modified. The four grid points at the corners of the stretched grid are filled with the data from the next neighboring grid point at the same height. For example, the wind speed at the corner point (-76m, 50m) is assumed identical to the point at (-76m, 25m).

Regarding the distortion of the eddies, we fully agree with your comment. This is addressed in L.233-237 in the manuscript. The stretching of the GROWIAN spatial arrangement is a simplification with limitations. The method can certainly be improved. However, for practical uses in our paper, we assume the validity of the self-similarity of turbulent flows, and hypothesize that typical wind structures on the scale of the WT are obtained through the simple stretching method.

**RESULTS**

The BEM results are low-pass filtered as CoWP is a good description of large-scale turbulent fluctuations. The signals are also zero-meaned and normalized to have a standard deviation of 1. In the context of developing a surrogate model the manipulations that are done to the data seem to be significant. What is the effect on the long-term statistics and extrapolated loads of the filtered-out high-frequency component?

Thank you for your question. The differences of the calculations between the low-pass filtered and the total signals have been a topic of discussion, also inquired by the other referee. The contribution of the low- and high-frequency components of the loads to the total signal were investigated. The main results of the analysis are now included in Appendix B in the manuscript.

As a summary, it was found first, that the high-frequency component has a minor contribution, compared to the low-frequency, to the DEL calculated from the complete load signal. The significance of each component is quantified by the factors $\alpha$ and $\beta$ in Eq. (B1). Second, the high-frequency signals can be assumed as noise with added autocorrelation and a dominant frequency. Then, it can be easily modeled by a simple stochastic process (see Fig. B2).

In response to your comment, and after the analysis of the high-frequency modeling, L.293-299 have been added.

It is essential to acknowledge that the discussion on the DELs presented in our work is exclusively focused on the DELs from the low-frequency component of the signals. This choice is based on a particular interest of our research

partners. In order to assess the complete DELs (e.g., from both the low- and high-frequency load events), it is necessary to establish an additional model for incorporating the contribution from the high-frequency signal. In this direction, a simple surrogate stochastic model has shown satisfactory results. The characteristics of the original high-frequency load signal are well reproduced. The proposed stochastic model for the high-frequency signals, and calculations on the differences between the DELs from the low- and high-frequency load components, and total DELs are shown in Appendix B.

Regarding the normalization of the signals – given the excellent statistical agreement between the normalized signal statistics, it would be interesting to see a transfer function mapping the CoWP to yaw bearing bending moments or other wind turbine load sensors as the author see fit.

Yes, we agree entirely with your suggestion. A transfer function between the calculated CoWP from the wind fields and the low-frequency component of the bending moments at the main shaft of the WT is the immediate next step of our method. Such a transfer function would be turbine-specific, as the magnitudes of the loads will depend on the structural properties of the WT.

Along with the analysis presented in this paper, such a transfer function corresponds to a simple linear scaling parametrized by the mean and standard deviation of the original load signals. However, the term transfer function remains general, as there may be more complex cases requiring more than a linear scaling.

The transfer function between the CoWP and the loads is one of the key elements for the applicability of the approach for load assessment of operating WTs. Although the formulation and validation of such transfer functions is out of the scope of this paper, a sentence, in L.267, has been added into the manuscript.

A turbine-specific transfer function for rescaling the normalized values of the CoWP to magnitudes of the low-frequency component of operational bending moments would be necessary for the assessment of the loads in engineering applications. Such a transfer function will therefore depend on the structural properties of the WT and particular control mechanisms.

The topic is also now discussed as part of the outlook in the manuscript (L.393-396).

However, the development of lifetime predictions in engineering applications necessitates the incorporation of additional elements in conjunction with the proposed stochastic method for modeling the low-frequency component of the loads. Initially, a turbine-specific transfer function for rescaling the CoWP to the magnitudes of the low-frequency component of the bending moments should be derived.

Figure 8: When commenting this figure I would highlight the fact that the DELS agree well in an aggregate sense, but less so on a simulation per simulation

perspective. Indeed, while statistics are in very good agreement (c, d) and correlation is good (a, b) a large spread in the data con be seen in figures 8 (a) and 8 (b).

Thank you very much for the suggestion. In fact, it is very important to make a comment on this. L. 289 has been added to the manuscript.

Overall, the data in Fig. 8 reveal a very good agreement between the DEL and $DEL_{CoWP}$ in a statistical sense. Although a spread of the data is observed, the statistics and correlation are consistent. In an aggregate sense, these results indicate an equivalence between the CoWP and the bending moments.

Finally, please provide more details on the BEM numerical setup. Some details are included in the provided reference but should be repeated herein since the simulations constitute the reference for the entire work.

Thank you for the request. The repetition of the information in this paper adds important details of the simulations. The paragraph, L.226-231, has been modified in the manuscript.

The multi-body model alaska/Wind incorporates several coupled sub-models: the foundation, the tower, the nacelle, the yaw drive, the pitch drive, the rotor, the drive train, the generator, and the controller. A Beddoes-Leishman-type dynamic model and a flexible wake model respectively consider the unsteady airfoil aerodynamics and the wake effects. Among the specified degrees of freedom are the radial degree in the drive train for torsional effects of the gearbox; the nodding degree in the yaw drive; and the side-to-side, fore-aft, and torsional motions of the tower. These considerations on the modeling assumptions of the simulations follow the simulations in [1].

**A new version of the manuscript is provided along with a diff file**.

**References**

[1] Schubert, C., Moreno, D., Schwarte, J., Friedrich, J., Wächter, M., Pokriefke, G., Radons, G., and Peinke, J.: Introduction of the Virtual Center of Wind Pressure for correlating large-scale turbulent structures and wind turbine loads, Wind Energy Sci. Discuss. [preprint], 2025, 1–19, https://doi.org/10.5194/wes-2025-28, *in review*, 2025.

---

## Author Response (AR2)

**Response to Referee 2**

From the center of wind pressure to loads on the wind turbine: A stochastic approach for the reconstruction of load signals

Referee's comment (RC) in blue Author's comment (AC) in black

The references to lines in the manuscript (e.g., 'L.80') are given with respect to the **new version** of the paper.

In gray: text from the revised version of the manuscript.

**GENERAL COMMENTS**

REFEREE:

[...] I feel there is still some confusion regarding the applicability of the method to fatigue or extreme loads. In fact, while evidence is provided on how the method may be applied to lifetime fatigue load estimation, it is not validated for extreme load prediction. I think most of the confusion lies in the fact that the authors sometimes refer to extreme loads caused by gusts, other times to extreme load extrapolation. While standards prescribe both scenarios to be checked, they are very different situations, and this is a bit confused in the way it is presented now. Also more explicitly distinguishing between fatigue and extreme loads in the text is recommended, see below for more details.

The comment on the fact that the method has been proposed for lifetime extreme load extrapolation has been only partially addressed. From the authors response throughout the paper and in the conclusion it is not clear if "Lifetime loads" are fatigue or extreme loads. The paper provides evidence that the method may indeed predict fatigue loads accurately, but evidence regarding extremes is not provided. Extreme lifetime loads (1yr or 50yr return period) are often difficult to predict with a simplified model as they may depend on a reduced number of cases. It is extremely difficult to predict which cases those may be due to the high non-linearity of the system. And thus discrepancies in these scenarios, which are often hard to tune for due to their scarcity, may cause differences in predictions.

I would thus recommend to clarify better if the discussion, and the procedure outlined in the conclusions to go from COWP to lifetime loads refers to fatigue or extreme loads. When referring to extreme loads I would recommend to either provide more evidence of the ability of the model to predict 50 or 1-year extreme loads or to clearly state that while the model may provide an interesting prospect in this regard, it is yet to be verified and validated.

Thank you very much for your comment. By re-reading our statements concerning extreme wind and extreme loads, we realize, as you remarked, that they might lead to ambiguity. However, it should be noted that in our work, we do not refer to extreme loads as defined by the IEC (i.e., with a 50-year or 1-year return period). Within the framework of the CoWP, the designation 'extreme load' has been adopted to signify substantial load events that, with a given wöhler coefficient, drive the estimated Damage Equivalent Loads (DELs) [1]. The importance of these very large events, which can be reproduced by our proposed stochastic method, is grounded on their dominant contribution to the DEL without being categorized within the scope of an IEC extreme load scenario.

To reduce possible confusion with the term 'extreme', our text has been modified in the manuscript.

In the paper, we employ the term 'extreme wind events' to denote specific instances of localized gust-like structures, which result in very strong differences of the wind speed over the y-z plane (i.e., perpendicular to the predominant direction of the wind, as illustrated in Fig. D1). In order to differentiate these events from the extreme wind conditions defined by the IEC, we have modified lines 337-339, 366-368, and 498. In such instances, the term 'extreme wind' has been replaced by 'severe or very strong differences of the wind speed over the rotor plane'. As an example, L.337-339:

Over those intervals, significant differences in the wind speed are observed in the spatial domain (i.e., over the rotor plane).

As an additional aspect, we want to point out that the dynamic response of modern wind turbines with increased size gives additional relevance to wind structures over the rotor plane. Larger areas covered by the increased rotors likely include inhomogeneities (i.e., severe differences in wind speed) over the rotor. In this direction, the CoWP and the stochastic approach delineated in our paper have the potential to serve as a tool for describing and modeling IEC extreme scenarios.

The following lines have been added to the conclusions of the manuscript (L. 426-429):

The validity of this load estimation has been demonstrated in the context of the DELs. The dynamic response of modern wind turbines with increased size gives additional relevance to wind structures over the rotor plane. Larger areas covered by the increased rotors likely include inhomogeneities (e.g., severe differences in wind speed) over the rotor. In this direction, the CoWP and the stochastic approach delineated in our paper have the potential to serve as a tool for describing and modeling IEC extreme scenarios (i.e., with 50-year or 1-year return period). In this direction, the CoWP and the stochastic approach delineated in our paper have the potential to serve as a tool for describing and modeling IEC extreme scenarios (i.e., with 50-year or 1-year return period).

**A new version of the manuscript is provided along with a diff file.**

[1] Schubert, C., Moreno, D., Schwarte, J., Friedrich, J., Wächter, M., Pokriefke, G., Radons, G., and Peinke, J.: Introduction of the Virtual Center of Wind Pressure for correlating large-scale turbulent structures and wind tur- bine loads, Wind Energy Sci. Discuss. [preprint], 2025, 1–19, https://doi.org/10.5194/wes-2025-28, in review, 2025.